# Robustness in deep learning: The good (width), the bad (depth), and the ugly (initialization)

**Zhenyu Zhu,    Fanghui Liu,    Grigorios G Chrysos,    Volkan Cevher**

EPFL, Switzerland
{[[first name].[surname]}@epfl.ch

## Abstract

We study the average robustness notion in deep neural networks in (selected) wide and narrow, deep and shallow, as well as lazy and non-lazy training settings. We prove that in the under-parameterized setting, width has a negative effect while it improves robustness in the over-parameterized setting. The effect of depth closely depends on the initialization and the training mode. In particular, when initialized with LeCun initialization, depth helps robustness with the lazy training regime. In contrast, when initialized with Neural Tangent Kernel (NTK) and He-initialization, depth hurts the robustness. Moreover, under the non-lazy training regime, we demonstrate how the width of a two-layer ReLU network benefits robustness. Our theoretical developments improve the results by Huang et al. [2021], Wu et al. [2021] and are consistent with Bubeck and Sellke [2021], Bubeck et al. [2021].

## 1   Introduction

It is now well-known that deep neural networks (DNNs) are susceptible to adversarially chosen, albeit imperceptible, perturbations to their inputs [Goodfellow et al., 2015, Szegedy et al., 2014]. This lack of robustness is worrying as DNNs are now deployed in many real-world applications [Eykholt et al., 2018]. As a result, new algorithms are more and more being developed to defend against adversarial attacks to improve the DNN robustness. Among the current defense methods, the most commonly used and arguably the most successful method is adversarial training based minimax optimization [Athalye et al., 2018, Croce and Hein, 2020, Madry et al., 2018]. To study adversarial attacks and defenses, we need to investigate the robustness of DNNs at first.

A plethora of aspects on the robustness have been studied, ranging from algorithms to their initialization as well as from the width of neural networks to their depth (i.e., the architecture). On the practical side, Madry et al. [2018] advocate that adversarial training requires more parameters (e.g., width) for better performance in minimax optimization, which would fall into the so-called over-parameterized regime[1] [Zhang et al., 2017]. On the theoretical side, recent works suggest that over-parameterization may damage the adversarial robustness [Huang et al., 2021, Wu et al., 2021, Zhou and Schoellig, 2019, Hassani and Javanmard, 2022]. In stark contrast, Bubeck and Sellke [2021], Bubeck et al. [2021] argue that the robustness of DNNs needs enough parameters to be guaranteed. See a detailed discussion in Section 2.

Our work aims to investigate this apparent contradiction in theory, and to close the gap as much as possible. We begin with a definition of the *perturbation stability* of DNNs, which can be used to describe the robustness, following the spirit of Wu et al. [2021], Dohmatob and Bietti [2022].

**Definition 1** (*perturbation stability*)**.** *The perturbation stability of a neural network $f(x; W)$ : $\mathbb{R}^d \mapsto \mathbb{R}^o$ parameterized by the neural network parameter $W$ under the data distribution $\mathcal{D}_X$ and a*

---

[1]Over-parameterized regime requires the number of parameters in DNN to be (much) larger than the number of training data.

36th Conference on Neural Information Processing Systems (NeurIPS 2022).

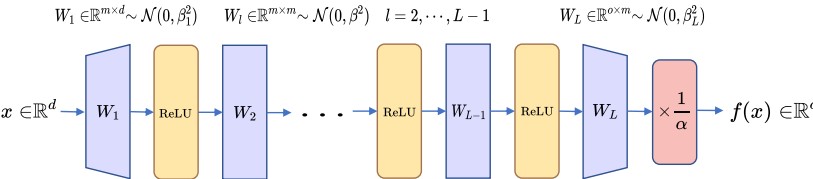

Figure 1: Schematic of our deep fully connected ReLU neural network.

Table 1: Comparison of the *perturbation stability* of a deep ReLU neural network (see Fig. 1) under three common Gaussian initializations with different variances. For a formal definition of this neural network please refer to Eq. (2).

| Initialization name | Initialization form | Our bound for $\mathscr{P}(\boldsymbol{f}, \epsilon)/\epsilon$ |
|---|---|---|
| LeCun et al. [2012] | $\beta_1 = \sqrt{\frac{1}{d}}, \beta = \beta_L = \sqrt{\frac{1}{m}}, \alpha = 1$ | $\left(\sqrt{\frac{L^3 m}{d}} e^{-m/L^3} + \sqrt{\frac{o}{d}}\right)(\frac{\sqrt{2}}{2})^{L-2}$ |
| He et al. [2015] | $\beta_1 = \sqrt{\frac{2}{d}}, \beta = \beta_L = \sqrt{\frac{2}{m}}, \alpha = 1$ | $\sqrt{\frac{L^3 m}{d}} e^{-m/L^3} + \sqrt{\frac{o}{d}}$ |
| Allen-Zhu et al. [2019] | $\beta = \beta_1 = \sqrt{\frac{2}{m}}, \beta_L = \sqrt{\frac{1}{o}}, \alpha = 1$ | $\sqrt{\frac{L^3 m}{o}} e^{-m/L^3} + 1$ |

*perturbation radius $\epsilon$ is defined as follows:*

$$\mathscr{P}(\boldsymbol{f}, \epsilon) = \mathbb{E}_{\boldsymbol{x}, \hat{\boldsymbol{x}}, \boldsymbol{W}} \left\| \nabla_{\boldsymbol{x}} \boldsymbol{f}(\boldsymbol{x}; \boldsymbol{W})^\top (\boldsymbol{x} - \hat{\boldsymbol{x}}) \right\|_2, \quad \forall \boldsymbol{x} \sim \mathcal{D}_X, \ \hat{\boldsymbol{x}} \sim Unif(\mathbb{B}(\epsilon, \boldsymbol{x})), \qquad (1)$$

*where $\hat{\boldsymbol{x}}$ is uniformly sampled from an $\ell_2$ norm ball of $\boldsymbol{x}$ with radius $\epsilon$, denoted as $Unif(\mathbb{B}(\epsilon, \boldsymbol{x}))$.*

Since our definition of the *perturbation stability* takes the expectation of the clean and the adversarial data points, it is natural to describe the *average robustness* of a neural network. It can be noticed that the larger value of the *perturbation stability* means worse robustness in average, i.e., *average robustness*.

Based on the *perturbation stability*, we study the *average robustness* of neural networks under different initializations in (selected) wide and narrow, deep and shallow, as well as lazy and non-lazy[2] training settings. Generally, non-lazy training makes the analysis of neural networks intractable as DNNs in this regime cannot be simplified as a time-independent model [Chizat and Bach, 2018], and accordingly, the analysis in this regime is mainly restricted to the two-layer setting [Mei et al., 2018, 2019].

Overall, our results suggest that the width (*good*) helps *average robustness* in the over-parameterized regime but the depth (*bad*) can help only under certain initializations (*ugly*). To be specific, we make the following contributions and findings under the lazy/non-lazy training regimes, see Table 1.

In the **lazy training** regime, the derived upper-bounds for DNNs, suggest that

- along with the increase in width, the robustness firstly becomes worse in the under-parameterized regime and then gets better, and finally tends to be a constant in highly over-parameterized regimes, which implies the existence of a phase transition.
- the depth has more complex tendency on robustness, which largely depends on the initialization and the training mode. It can be grouped into two main classes (*cf.*, Table 1): depth helps robustness in an exponential order under the LeCun initialization [LeCun et al., 2012], whereas it hurts robustness in a polynomial order under He-initialization [He et al., 2015] and under Neural Tangent Kernel (NTK) initialization [Allen-Zhu et al., 2019].

Surprisingly, standard tools on training dynamics of neural networks [Allen-Zhu et al., 2019, Du et al., 2018] are sufficient to obtain our bounds, which explain the relationship between robustness

---

[2]Here the lazy/non-lazy training regime indicates that neural network parameters change little/much during training. These two phases are determined by different initializations [Woodworth et al., 2020, Luo et al., 2021].

and the structural/architectural parameters of neural network. Our theoretical developments improve the results by Huang et al. [2021], Wu et al. [2021], and are supported by empirical evidence.

In the **non-lazy** training regime, we derive upper-bounds for two-layer networks, suggesting that

- the width improves the robustness under different initializations.
- the convergence rates of the average robustness is affected by the initialization.

We also derive a sufficient condition to identify when DNNs enter in this regime, as an initial but first attempt on understanding DNNs in this regime. Our technical contribution lies in connecting robustness to changes of neural network parameters during the early stages of training, which could expand the application scope of deep learning theory beyond *lazy training* analysis [Jacot et al., 2018, Allen-Zhu et al., 2019].

**Notations:** We use the shorthand $[n] := \{1, 2, \ldots, n\}$ for a positive integer $n$. We denote by $a(n) \lesssim b(n)$: there exists a positive constant $c$ independent of $n$ such that $a(n) \leqslant cb(n)$. The standard Gaussian distribution is $\mathcal{N}(0, 1)$ with the zero-mean and the identity variance. Uniform distribution inside the sphere is $\mathrm{Unif}(\mathbb{B}(\epsilon, \boldsymbol{x}))$ with the center $\boldsymbol{x}$ and radius $\epsilon$. We follow the standard Bachmann–Landau notation in complexity theory e.g., $\mathcal{O}, o, \Omega$, and $\Theta$ for order notation.

## 2 Related work

DNNs are demonstrated to be sensitive to adversarially chosen but undetectable noise both empirically [Szegedy et al., 2014] and theoretically [Huang et al., 2021, Bubeck and Sellke, 2021]. Adversarial training [Athalye et al., 2018, Croce and Hein, 2020, Zhang et al., 2020b] is a reliable way to obtain adversarially robust neural network. Nevertheless, improving the overall robustness of neural networks is still an unsolved problem in machine learning, especially when coupling with initializations and parameters.

**Over-parameterized neural networks under lazy/non-lazy training regimes:** Modern DNNs in practice [He et al., 2016] work under the setting where the number of parameters is (much) larger than the number of training data. Analysis of DNNs in terms of optimization [Safran et al., 2021, Zhou et al., 2021] and generalization [Cao and Gu, 2019] has received great attention in deep learning theory [Zhang et al., 2017].

In deep learning theory, neural tangent kernel (NTK) [Jacot et al., 2018] and mean field [Mei et al., 2018] analysis are two powerful tools for neural network analysis. To be specific, NTK builds an equivalence between training dynamics by gradient-based algorithms of DNNs and kernel regression under a specific initialization, and thus allows the analysis of deep networks [Allen-Zhu et al., 2019, Du et al., 2019a, Chen et al., 2020]. However, the NTK requires neural networks to belong in the *lazy training* regime [Chizat et al., 2019], where neural networks are able to achieve zero training loss but the parameters change little, or even remain unchanged during training. In contrast, mean-field theory establishes global convergence by casting the network weights during training as an evolution in the space of probability distributions under some certain initializations [Mei et al., 2018, Chizat and Bach, 2018]. This strategy goes beyond lazy training regime, which allows for neural networks parameters to change in a constant order after training.

If the neural networks parameters change a lot after training, or even tend to infinity, then neural networks work in the *non-lazy* training regime. Analysis of DNNs under this setting appears intractable and challenging, so the current work mainly focus as two-layer neural networks [Maennel et al., 2018, Luo et al., 2021].

**Robustness and over-parameterization** Goodfellow et al. [2015] demonstrate that adversarial learning helps robustness and reduces overfitting. Many works focus on influencing factors of adversarial examples and robustness of the neural network [Schmidt et al., 2018, Zhang et al., 2020a, Allen-Zhu and Li, 2022]. The relation between model capacity and robustness is empirically investigated by Madry et al. [2018], i.e., a neural network with insufficient capacity can seriously hurt the robustness. Bubeck et al. [2021] theoretically study the inherent trade-off between the size of neural networks and their robustness, and they claim that over-parameterization is necessary for the robustness of two-layer neural networks.

However, some recent works propose the opposite view. Under the lazy training regime, Huang et al. [2021] demonstrate that when over-parameterized neural networks get wider, the robustness

decreases in a polynomial order. Similarly, the depth hurts the robustness in an exponential order. Wu et al. [2021] affirm the view of Huang et al. [2021] on the width. However for depth, they derive a stronger bound that the robustness gets worse in a polynomial decay as the depth increases, as suggested by Hassani and Javanmard [2022]: over-parameterization hurts robustness. In addition, Gao et al. [2019] also make a similar view: an increased model capacity (i.e., wider width and deeper depth) deteriorates the robustness of neural networks. Nevertheless, we remark that, the results of Hassani and Javanmard [2022] work in a slightly different setting than that of Bubeck et al. [2021] on data interpolation, which requires a careful comparison. Accordingly, we adopt a complementary view to the vast literature. We provide an in-depth theoretical analysis to investigate this apparent contradiction in theory, and to close the gap as much as possible.

## 3 Problem setting

Let $X \subseteq \mathbb{R}^d$ and $Y \subseteq \mathbb{R}^o$ be compact metric spaces. We assume that the training set $\mathcal{D}_{tr} = \{(\boldsymbol{x}_i, \boldsymbol{y}_i)\}_{i=1}^n$ is drawn from a unknown probability measure $\mathcal{D}$ on $X \times Y$. Its marginal data distribution is denoted by $\mathcal{D}_X$. The goal of the classification task is to learn a neural network $\boldsymbol{f} : X \to Y$ such that $\boldsymbol{f}(\boldsymbol{x}; \boldsymbol{W})$ parameterized by $\boldsymbol{W}$ is a good approximation of the label $\boldsymbol{y} \in Y$ corresponding to a new sample $\boldsymbol{x} \in X$. In this paper, we use the empirical risk $L(\boldsymbol{W}) = \frac{1}{2n} \sum_{i=1}^n \|\boldsymbol{f}(\boldsymbol{x}_i; \boldsymbol{W}) - \boldsymbol{y}_i\|_2^2$. Then we make the following assumption.

**Assumption 1.** *We assume that the data satisfy $\|\boldsymbol{x}\|_2 = 1$.*

**Remark:** This assumption is standard in theory of over-parameterized neural networks and also commonly used in practice [Du et al., 2019b,a, Allen-Zhu et al., 2019, Oymak and Soltanolkotabi, 2020, Malach et al., 2020].

### 3.1 Network

We focus on the typical depth-$L$ fully-connected ReLU neural networks admitting the width $m_l$ of the $l$-th hidden layer, $\forall l \in [L]$ (*cf.*, Fig. 1):

$$\boldsymbol{h}_{i,0} = \boldsymbol{x}_i; \quad \boldsymbol{h}_{i,l} = \phi(\boldsymbol{W}_l \boldsymbol{h}_{i,l-1}); \quad \boldsymbol{f}(\boldsymbol{x}_i; \boldsymbol{W}) = \boldsymbol{f}_i = \frac{1}{\alpha} \boldsymbol{W}_L \boldsymbol{h}_{i,L-1}; \quad \forall l \in [L-1], \ i \in [n], \tag{2}$$

where $\boldsymbol{x} \in \mathbb{R}^d$, $\boldsymbol{f}(\boldsymbol{x}) \in \mathbb{R}^o$, $\alpha$ is the scaling factor, and $\phi = \max(0, x)$ is the ReLU activation function. The neural network parameters formulate the tuple of weight matrices $\boldsymbol{W} := \{\boldsymbol{W}_i\}_{i=1}^L \in \{\mathbb{R}^{m \times d} \times (\mathbb{R}^{m \times m})^{L-2} \times \mathbb{R}^{o \times m}\}$. According to the property $\phi(x) = x\phi'(x)$ of ReLU, we have $\boldsymbol{h}_{i,l} = \boldsymbol{D}_{i,l} \boldsymbol{W}_l \boldsymbol{h}_{i,l-1}$, where $\boldsymbol{D}_{i,l}$ is a diagonal matrix under the ReLU activation function defined as below.

**Definition 2** (Diagonal sign matrix). *For each $i \in [n]$, $l \in [L-1]$ and $k \in [m]$, the diagonal sign matrix $\boldsymbol{D}_{i,l}$ is defined as: $(\boldsymbol{D}_{i,l})_{k,k} = \mathbb{1}\{(\boldsymbol{W}_l \boldsymbol{h}_{i,l-1})_k \geq 0\}$.*

In our setting, we consider the standard Gaussian initialization with different variances that includes three typical initialization schemes in practice.

**Initialization:** Let $m_0 = d$, $m_L = o$ and $m_2 = \cdots = m_{L-1} = m$, we make the standard random initialization $[\boldsymbol{W}_l]_{i,j} \sim \mathcal{N}(0, \beta_l^2)$ for every $(i, j) \in [m_l] \times [m_{l-1}]$ and $l \in [L]$. Choosing a different variance, our work holds for three commonly used Gaussian initializations, i.e., LeCun initialization [LeCun et al., 2012], He-initialization [He et al., 2015] and Neural Tangent Kernel (NTK) initialization [Allen-Zhu et al., 2019], refer to the formal definition in Table 1 for details.

### 3.2 Discussion on various robustness metrics

In Section 1, we have proposed our robustness metric: perturbation stability (*cf.*, Definition 1). This metric can be viewed as an expectation of the inner product of first-order approximation of adversarial risk [Madry et al., 2018] and the perturbations with the uniform distribution, which measures the *average robustness* of the neural network. As we mentioned before, under the same perturbation radius $\epsilon$, a smaller value of $\mathcal{P}(\boldsymbol{f}, \epsilon)$ implies better *perturbation stability*, that is better *average robustness*. Previous works [Hein and Andriushchenko, 2017, Weng et al., 2018, Wu et al., 2021, Bubeck and Sellke, 2021] use Lipschitzness to describe the robustness of the network, suggesting that smaller Lipschitzness leads to robust models. However, Lipschitzness is only a worst-case measure, and cannot reasonably describe the average changes of the entire dataset. Instead, we follow the

measure of Wu et al. [2021], Dohmatob and Bietti [2022], that aims to comprehensively consider the overall distribution of the data, not only the extreme case. Besides, the worst-case robustness can be extended to a probabilistic robustness view [Robey et al., 2022], which shares a similar spirit as our average robustness concept. Schmidt et al. [2018] present another definition of robustness, depending on the misclassified error of an adversarial data point. Instead, our perturbation stability measures the function value changes at the clean data point via Taylor expansion which can exclude the influence of the learning capacity of the network.

# 4 Main results

In this section, we state the main theoretical results. Firstly, in Section 4.1 we provide the upper bound of the *perturbation stability* in lazy training regime for deep neural networks defined by Eq. (2). The sufficient condition that the neural network Eq. (2) is under non-lazy training regime is given in Section 4.2. Finally, in Section 4.3, we provide the upper bound on the *perturbation stability* during early training of two-layer network under the non-lazy training regime.

## 4.1 Upper bound of the perturbation stability of DNNs under the lazy training regime

We are now ready to state the main results under the lazy training regime. The following theorem provides the upper bound of the *perturbation stability* and connects to the width, and the depth of a deep fully-connected neural network under different standard Gaussian initializations. The proof of Theorem 1 is deferred to Appendix B.

**Theorem 1.** *Given an $L$-layer neural network $\boldsymbol{f}$ defined by Eq. (2) trained by $\{(\boldsymbol{x}_i, \boldsymbol{y}_i)\}_{i=1}^n$ satisfying Assumption 1, for the convenience of analysis, we set $\alpha = 1$ and $\beta := \beta_2 = \cdots = \beta_{L-1}$, define a constant $\gamma := \beta / \sqrt{\frac{2}{m}}$. Then, under a small perturbation $\epsilon$, we have the following:*

$$\frac{\mathscr{P}(\boldsymbol{f}, \epsilon)}{\epsilon} \lesssim \left( \sqrt{L^3 m^2 \beta_1^2 \beta_L^2} e^{-m/L^3} + \sqrt{m o \beta_1^2 \beta_L^2} \right) \gamma^{L-2} . \tag{3}$$

**Remark:** Our results cover the effect of the width and depth of neural network on robustness under various common initializations depending on $\gamma \gtreqqless 1$.

**Comparison with three commonly used initializations:** For the initializations used in practice, our theoretical results can be mainly divided into two classes: 1) The depth helps robustness in an exponential order under the LeCun initialization: Theorem 1 implies that $\left( \sqrt{\frac{L^3 m}{d}} e^{-m/L^3} + \sqrt{\frac{o}{d}} \right) (\frac{\sqrt{2}}{2})^{L-2}$. 2) The depth hurts the robustness in a polynomial order under He-initialization $\left( \sqrt{\frac{L^3 m}{d}} e^{-m/L^3} + \sqrt{\frac{o}{d}} \right)$ and under the NTK initialization $\left( \sqrt{\frac{L^3 m}{o}} e^{-m/L^3} + 1 \right)$ derived by Theorem 1. When employing other initializations, the robustness could be hurted in a exponential order. Below, we elaborate on these three initalizations:

**1) LeCun initialization** ($\gamma = \frac{\sqrt{2}}{2}$): The order has three main parts: $\sqrt{\frac{L^3 m}{d}}$, $e^{-m/L^3}$ and $(\frac{\sqrt{2}}{2})^{L-2}$. Regarding the width $m$, the first part $\sqrt{\frac{L^3 m}{d}}$ is an increasing function of $m$ and the second part $e^{-m/L^3}$ is a decreasing function of $m$. Accordingly, in the under-parameterized region (e.g., $m$ is small), $\sqrt{\frac{L^3 m}{d}}$ plays a major role, so the stability will increase as $m$ increases. After a critical point, $e^{-m/L^3}$ plays a major role, so the stability will decrease as $m$ increases. When $m$ tends to infinity, the first term of the bound tends to $0$. Hence the *perturbation stability* tends to be a constant and independent of the width $m$ as the width $m$ tends to infinity. It means that there exists a phase transition phenomenon between the *perturbation stability* and over-parameterization in terms of the width $m$.

Regarding the depth $L$, the constant $\gamma = \frac{\sqrt{2}}{2}$ implies that the third part has a faster decrease speed than the first and second parts and plays a major role in the tendency. The *perturbation stability* of the neural network exponentially decreases with respect to the depth. That means, for the LeCun initialization, the deeper the network, the better the robustness. Nevertheless, the energy of the LeCun

Table 2: Comparison of the orders of the proposed bound with other two recent works. Our results are general to cover both under- and over-parameterized regimes, which expands the application scope of previous results [Wu et al., 2021, Huang et al., 2021]. (The original result of [Wu et al., 2021] can be reduced to $\sqrt{mL}$ as the $\frac{m}{(\log m)^6} \geq L^{12}$ condition is required).

| Metrics | Our result | Wu et al. [2021] | Huang et al. [2021] |
|---|---|---|---|
| $\mathscr{P}(\boldsymbol{f}, \epsilon)/\epsilon$ | $\sqrt{\frac{L^3 m}{o}} e^{-m/L^3} + 1$ | $L^2 m^{1/3} \sqrt{\log m} + \sqrt{mL}$ | $2^{\frac{3L-5}{2}} \sqrt{m}$ |

initialization decreases as the network depth increases due to the variance $\beta = \sqrt{\frac{1}{m}}$. Since the activation function ReLU loses half of the energy in every layer, training a deep network with the LeCun initialization is difficult. Hence, we need a trade off between robustness and training difficulty regarding the network depth in practice for the LeCun initialization.

**2) He initialization and NTK initialization** ($\gamma = 1$): the bounds for these two initializations are almost the same, and only differ in the feature dimension. We can see that phase transition phenomena exist under these two initializations regarding the width $m$, similar to the LeCun initialization. Regarding the depth, when $L$ is large, the first part $\sqrt{L^3}$ plays a major role in the *perturbation stability*. So these two initializations hurt the robustness of the neural network at a polynomial order.

All of the three initializations admit $\gamma \leq 1$. If some initialization schemes admit $\gamma > 1$, then the depth $L$ will hurt the robustness of the neural network at an exponentially increasing rate.

**Comparison with previous work:** Theorem 1 provides a new relationship between the robustness with width and depth of DNNs. We compare our result with [Wu et al., 2021, Huang et al., 2021] using a basic NTK initialization [Allen-Zhu et al., 2019] (suppose that $m \gg o$ and $m \gg d$). For a better comparison, we derive their results under our robustness metric *perturbation stability*, as reported by Table 2.

Our results indicate a behavior transition on the width. For the over-parameterized regime, the robustness of the neural network only depends on the perturbation energy, and it is almost independent of the width $m$. The results on the width are significantly better than the previous results increasing as the square root of $m$. For depth $L$, our results provide a tighter and more precise estimate as compared to [Wu et al., 2021] in a two-degree polynomial increasing order and [Huang et al., 2021] in an exponential increasing order.

Furthermore, compared with the results of Bubeck et al. [2021] showing that the robustness of the two-layer neural network becomes better with the increase of the number of neurons (i.e., width), we provide a more detailed and refined result on the robustness of DNNs under different initializations and under-parameterized regimes.

### 4.2 Sufficient condition for neural network under non-lazy training regime

Beyond the lazy training regime, we turn our attention to the non-lazy training regime and provide a sufficient condition for a (well-chosen) initialization of neural networks when entering into the non-lazy training regime. This is a first attempt to understand the training dynamics of DNNs in this regime.

Our result requires a further assumption on the data and the empirical risk as follows.

**Assumption 2.** *For a single-output network defined in Eq. (2), we assume that* $\max_{i \in [n]} y_i \geq C_1 > 0$ *for some constant* $C_1$. *We also assume that the neural network can be well-trained such that the empirical risk is* $\mathcal{O}(\frac{1}{n})$.

**Remark:** This is a common assumption in the field of optimization [Song et al., 2021] in the under- and over-parameterized regime, and we can even assume zero risk. Here we follow the specific assumption of Luo et al. [2021].

Now we are ready to present our result: a sufficient condition to identify when deep ReLU neural networks fall into the non-lazy training regime, as a promising extension of Luo et al. [2021] on two-layer neural networks. To avoid cluttering the analysis, we assume a single-output i.e., $o = 1$. The proof of Theorem 2 is deferred to Appendix C.

**Theorem 2.** *Given an L-layer neural network $\boldsymbol{f}$ defined by Eq. (2) with $o = 1$, trained by $\{(\boldsymbol{x}_i, \boldsymbol{y}_i)\}_{i=1}^n$, under Assumptions 1 and 2, suppose that $\alpha \gg (m^{3/2} \sum_{i=1}^L \beta_i)^L$ and $m \gg d$ in Eq. (2), then for sufficiently large $m$, with probability at least $1 - (L - 2) \exp(-\Theta(m^2)) - \exp(-\Theta(md)) - \exp(-\Theta(m))$ over the initialization, we have:*

$$\sup_{t \in [0, +\infty)} \frac{\|\boldsymbol{W}_l(t) - \boldsymbol{W}_l(0)\|_{\mathrm{F}}}{\|\boldsymbol{W}_l(0)\|_{\mathrm{F}}} \gg 1, \quad \forall l \in [L].$$

**Remark:** The condition $\alpha \gg (m^{3/2} \sum_{i=1}^L \beta_i)^L$ implies that, a neural network falls in a non-lazy training regime when the variance of the Gaussian initialization is very small. It can be achieved by a typical case: taking $m \gg L^2$, choosing $\alpha = 1$ and $\forall l \in [L]$; $\beta_l = \frac{1}{m^2}$. Commonly used initializations such as NTK initialization, LeCun initialization, He's initialization lead to lazy training.

### 4.3 Upper bound of the perturbation stability for two-layer networks in non-lazy training

Unlike lazy training, weights of non-lazy training concentrate on few directions determined by the input data in the early stages of training [Luo et al., 2021]. The following theorem describes the neural network *perturbation stability* in the early training stage as a function of network width in the non-lazy training regime. For ease of description, here we consider a special initialization scheme under the non-lazy regime, the proof of Theorem 3 is deferred to Appendix D.

**Theorem 3.** *Given a two-layer neural network with single output $f_t$ defined by Eq. (2) and trained by $\{(\boldsymbol{x}_i, y_i)\}_{i=1}^n$ satisfying Assumption 1, using gradient descent under the squared loss, consider the following initialization in Eq. (2) : $L = 2$, $\alpha \sim 1$, $\beta_1 \sim \beta_2 \sim \beta \sim \frac{1}{m^c}$ with $c \geq 1.5$, $m \gg n^2$ and training time less than a constant that only depends on $n, m$ and $\lambda_0$, then for a small range of perturbation $\epsilon$, with probability at least $1 - n \exp(-\frac{n}{2}) - \frac{3}{n}$ over initialization, we have the following:*

$$\frac{\mathscr{P}(f_t, \epsilon)}{\epsilon} \leq \Theta\left( \frac{\sqrt{n \log m} + n}{m^{c-1}} \left( \frac{1}{\sqrt{n^3 m}} + \frac{1}{m^{c-0.5}} \right) \right). \tag{4}$$

**Remark:** Under this setting of non-lazy training regime, the robustness and width of the neural network are positively correlated in the early stages of training. That is, as the width $m$ increases in the over-parameterized regime, a Gaussian initialization with smaller variance leads to the robustness increasing in a faster decay. Our result holds for other initialization schemes in the non-lazy training regime, e.g., $c = 2$ leads to $\mathscr{P}(f_t, \epsilon)/\epsilon \leq \Theta\left( \frac{\sqrt{n \log m} + n}{m^{2.5}} \right)$; and $c = 3$ leads to $\mathscr{P}(f_t, \epsilon)/\epsilon \leq \Theta\left( \frac{\sqrt{n \log m} + n}{m^{4.5}} \right)$.

## 5 Numerical evidence

We validate our theoretical results with a series of experiments. In Section 5.2 we firstly verify that our initialization settings belong in the lazy and the non-lazy training regimes. In Section 5.3, we explore the effect of varying widths from under-parameterized to over-parameterized regions on the *perturbation stability* of neural networks. In Section 5.4, we finally compare the effect of two different initializations and the network depth on the *perturbation stability*. Additional experimental results can be found in Appendix E.

### 5.1 Experimental settings

Here we present our experimental setting including models, hyper-parameters, the choice of width and depth, and initialization schemes. We use the popular datasets of MNIST [Lecun et al., 1998] and CIFAR-10 [Krizhevsky et al., 2014] for experimental validation.

**Models:** We report results using the following models: fully connected ReLU neural network named "FCN" in main paper and convolutional ReLU neural network named "CNN" in Appendix E.

**Hyper-parameters:** Unless mentioned otherwise, all models are trained for 50 epochs with a batch size of 64. The initial value of the learning rate is 0.001. After the first 25 epochs, the learning rate is

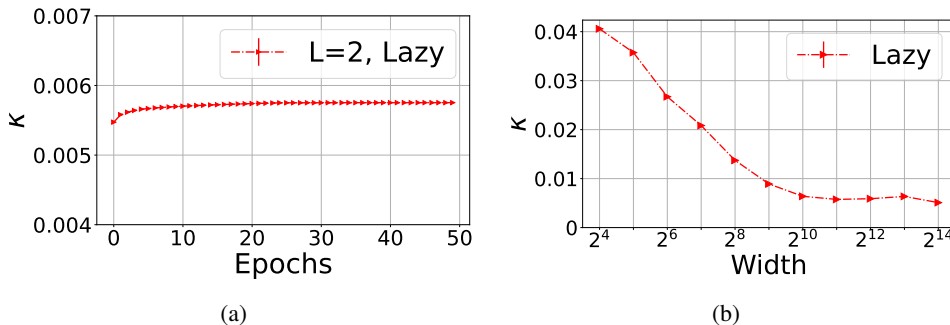

(a)                                                    (b)

Figure 2: (a) Tendency with respect to time (training epochs) and (b) relationship between width and lazy training ratio of neural networks. Fig. 2(a) shows that ratio $\kappa$ is small and almost unchanged, recognized as *lazy training*. In Fig. 2(b), we can see that the $\kappa$ decreases with the increasing width.

multiplied by a factor of $0.1$ every $10$ epochs. The SGD is used to optimize all the models, while the cross-entropy loss is used.

**Width and depth:** In order to verify our theoretical results, we conduct a series of experiments with different depths and widths of the same type neural network. Specifically, our experiments include 11 different widths from $2^4$ to $2^{14}$, and four different choices of depths, i.e., $2, 4, 6, 8, 10$.

**Initialization:** We report results using the following initializations: 1) He initialization where $W_{ij} \sim \mathcal{N}(0, \frac{2}{m_{in}})$, 2) LeCun initialization where $W_{ij} \sim \mathcal{N}(0, \frac{1}{m_{in}})$ and 3) an initialization that allows for non-lazy training regime on two-layer networks, i.e., $\beta_1 = \beta_2 = 1/m^2$ and $\alpha = 1$.

## 5.2 Validation of lazy and non-lazy training regimes

Before verifying our results, we need to identify the lazy and non-lazy training regime under different initializations. To this end, we define a measure for the *lazy training ratio*, i.e., $\kappa = \frac{\sum_{l=1}^{L} \|\boldsymbol{W}_l(t) - \boldsymbol{W}_l(0)\|_{\mathrm{F}}}{\sum_{l=1}^{L} \|\boldsymbol{W}_l(0)\|_{\mathrm{F}}}$. This measure evaluates whether the neural network is under the lazy training regime. A smaller $\kappa$ implies that the neural network is close to lazy training.

According to the theory, we employ the He initialization and the non lazy training initialization we state in Section 5.1 to conduct the experiment under two-layer neural networks to verify that their lazy training ratio matches the theoretical results of lazy training and non-lazy training (i.e., the experiment is under the correct regime). Fig. 2(a) and Fig. 2(b) show the tendency of ratio with respect to time (training epochs) and relationship between width and lazy training ratio of neural networks under lazy training regime, respectively. We find that the ratio of lazy traing regime is almost a constant that does not change with time, and this constant decreases as the width of the network increases. This is in line with what we know about lazy training [Chizat et al., 2019].

Likewise, Fig. 3 shows the ratio tendency with time and width under non-lazy training regime. The ratio increases almost linearly over time in Fig. 3(a). In epoch 25 we decrease the learning rate, which decreases the rate that $\kappa$ increases. At the same time, Fig. 3(b) shows a similar tendency between the width and lazy training ratio as lazy training. However, the value of $\kappa$ is much higher than that of lazy training regime. Combining the results about tendency with time, the ratio will be expected to increase as the time until infinity.

## 5.3 Validation for width

We verify the relationship between the *perturbation stability* and the width of network as illustrated by Eqs. (3) and (4). We conduct a series of experiments on MNIST dataset using FCN with different widths. Fig. 4 shows the relationship between the *perturbation stability* and width of FCN with different depths and training regimes. Here for lazy training and non-lazy training we use the same initialization as Section 5.2.

Fig. 4(a) exhibits the relationship between the *perturbation stability* and the width of neural networks with different depths for $L = 2, 4, 6, 8$, and $10$. All of the five curves confirm the phase transition with width: the *perturbation stability* firstly increases and then decreases with width, which match our

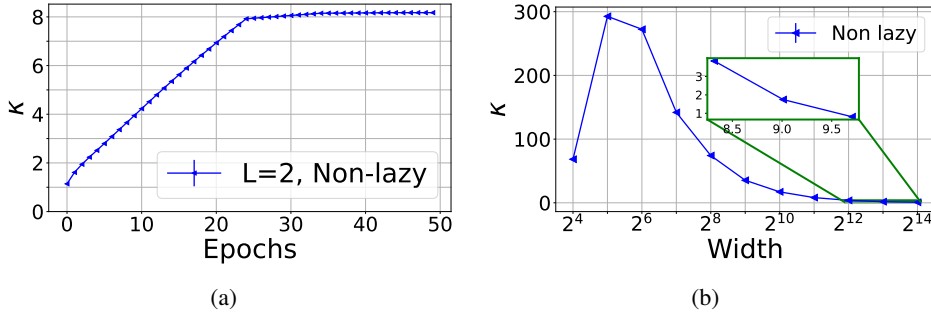

| (a) | (b) |
|:---:|:---:|

Figure 3: (a) Tendency with respect to time (training epochs) and relationship with width of non-lazy training ratio of neural networks. Fig. 3(a) shows that ratio $\kappa$ is changed a lot (increasing and then remains unchanged), recognized as *non-lazy training*. The tendency of $\kappa$ for non-lazy training is increasing with the width and then decreasing, i.e., a phase transition in Fig. 3(b).

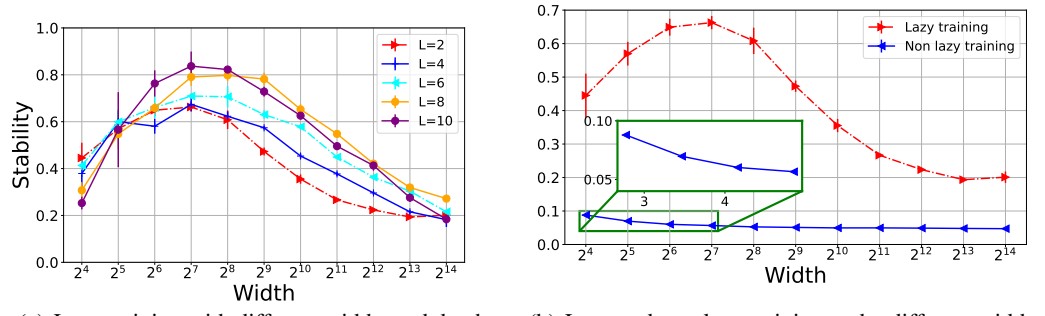

(a) Lazy training with different widths and depths     (b) Lazy and non-lazy training under different widths

Figure 4: Influence of width of neural network on the *perturbation stability*. (a) phase transition of the *perturbation stability vs.* width with five different depths under lazy training. (b) the difference between lazy training and non-lazy training regimes for two layer neural networks.

theoretical results. Fig. 4(b) shows the difference of the effect of width on the *perturbation stability* of lazy and non-lazy training for two-layer neural networks. The *perturbation stability* of non-lazy training is significantly smaller than that of lazy training, which means non-lazy training regime is more robust. Besides, the *perturbation stability* of non-lazy training decreases with the width of the neural network increases, which coincides with our theoretical result, i.e., no phase transition phenomenon.

### 5.4 Validation for depth and initialization

Let us explore the effect of depth on the *perturbation stability* under lazy training regime with different initializations in Fig. 5(a) and Fig. 5(b). Our results show the tendency of the *perturbation stability* for FCN with different widths and depths under the He initialization and the LeCun initialization, respectively. We observe a similar phase transition phenomenon, and find that, the *perturbation stability* under He initialization increases with depth, while the LeCun initialization shows the opposite tendency, which verified our theory.

## 6 Conclusions

In this work, we explore the interplay of the width, the depth and the initialization of neural networks on their average robustness with new theoretical bounds in an effort to address the apparent contradiction in the literature. Our theoretical results hold in both the under- and over-parameterized regimes. Intriguingly, we find a change of behavior in average robustness with respect to the depth, initially exacerbating robustness and then alleviating it. We suspect that this could help explain the contradictory messages in the literature. We also characterize the average robustness in the non-lazy training regime for two layer neural networks and find that width always help, coinciding with the results [Bubeck

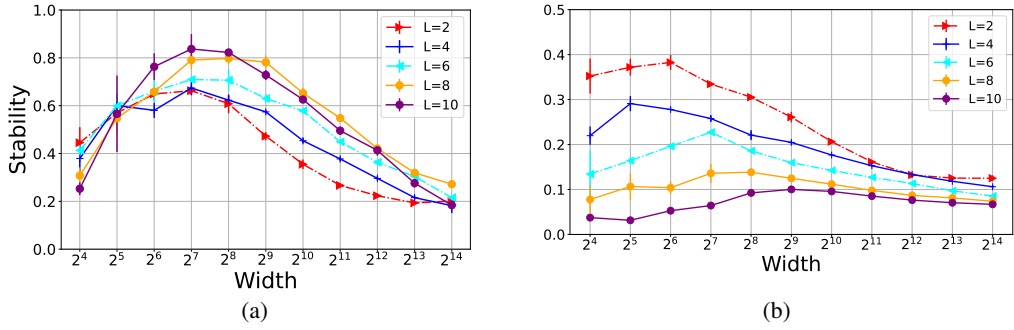

Figure 5: Relationship between the *perturbation stability* and depth of FCN under the He initialization (a) and the LeCun initialization (b) different depths of $L = 2, 4, 6, 8$ and $10$.

and Sellke, 2021, Bubeck et al., 2021]. We also provide numerical evidence to support the theoretical developments. Our results demonstrate the rationale behind robustness of over-parameterized neural networks and might be beneficial to analyze the highly over-parameterized foundation models, which have demonstrated exceptional performance in zero-shot performance [Brown et al., 2020].

## Acknowledgements

We are also thankful to the reviewers for providing constructive feedback. Research was sponsored by the Army Research Office and was accomplished under Grant Number W911NF-19-1-0404. This work was supported by Hasler Foundation Program: Hasler Responsible AI (project number 21043). This work was supported by SNF project - Deep Optimisation of the Swiss National Science Foundation (SNSF) under grant number 200021_205011. This work was supported by Zeiss. This project has received funding from the European Research Council (ERC) under the European Union's Horizon 2020 research and innovation programme (grant agreement n° 725594 - time-data). Corresponding authors: Fanghui Liu and Zhenyu Zhu.

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
