## Appendix introduction

The Appendix is organized as follows:

- In Appendix A, we state the symbols and notation used in this paper.
- In Appendix B, we provide the proofs and related lemmas of Theorem 1.
- In Appendix C, we provide the proofs of Theorem 2.
- In Appendix D, we provide the proofs and related lemmas of Theorem 3.
- In Appendix E, we detail our experimental settings and exhibit additional experimental results.
- In Appendix F, we discuss several limitations of this work.
- Finally, in Appendix G, we discuss the societal impact of this paper.

## A   Symbols and Notation

In the paper, vectors are indicated with bold small letters, matrices with bold capital letters. To facilitate the understanding of our work, we include the some core symbols and notation in Table 3.

Table 3: Core symbols and notations used in this project.

| Symbol | Dimension(s) | Definition |
|---|---|---|
| $\mathcal{N}(\mu, \sigma)$ | - | Gaussian distribution of mean $\mu$ and variance $\sigma$ |
| $\mathrm{Ber}(m, p)$ | - | Bernoulli (Binomial) distribution with $m$ trials and $p$ success rate. |
| $\chi^2(\omega)$ | - | Chi-square distribution of degree $\omega$. |
| $\|\boldsymbol{v}\|_2$ | - | Euclidean norms of vectors $\boldsymbol{v}$ |
| $\|\boldsymbol{M}\|_2$ | - | Spectral norms of matrices $\boldsymbol{M}$ |
| $\|\boldsymbol{M}\|_{\mathrm{F}}$ | - | Frobenius norms of matrices $\boldsymbol{M}$ |
| $\|\boldsymbol{M}\|_*$ | - | Nuclear norms of matrices $\boldsymbol{M}$ |
| $\lambda(\boldsymbol{M})$ | - | Eigenvalues of matrices $\boldsymbol{M}$ |
| $\boldsymbol{M}^{[l]}$ | - | $l$-th row of matrices $\boldsymbol{M}$ |
| $\boldsymbol{M}_{i,j}$ | - | $(i, j)$-th element of matrices $\boldsymbol{M}$ |
| $\phi(x) = \max(0, x)$ | - | ReLU activation function for scalar |
| $\phi(\boldsymbol{v}) = (\phi(v_1), \ldots, \phi(v_m))$ | - | ReLU activation function for vectors |
| $1\{A\}$ | - | Indicator function for event $A$ |
| $n$ | - | Size of the dataset |
| $d$ | - | Input size of the network |
| $o$ | - | Output size of the network |
| $L$ | - | Depth of the network |
| $m$ | - | Width of intermediate layer |
| $\beta_l$ | - | Standard deviation of Gaussian initialization of $l$-th intermediate layer |
| $\alpha$ | - | Scale factor for the output layer |
| $\boldsymbol{x}_i$ | $\mathbb{R}^d$ | The $i$-th data point |
| $\boldsymbol{y}_i$ | $\mathbb{R}^o$ | The $i$-th target vector |
| $\mathcal{D}_X$ | - | Input data distribution |
| $\mathcal{D}_Y$ | - | Target data distribution |
| $\boldsymbol{W}_1$ | $\mathbb{R}^{m \times d}$ | Weight matrix for the input layer |
| $\boldsymbol{W}_l$ | $\mathbb{R}^{m \times m}$ | Weight matrix for the $l$-th hidden layer |
| $\boldsymbol{W}_L$ | $\mathbb{R}^{o \times m}$ | Weight matrix for the output layer |
| $\boldsymbol{h}_{i,l}$ | $\mathbb{R}^m$ | The $l$-th layer activation for input $\boldsymbol{x}_i$ |
| $\boldsymbol{f}_i$ | $\mathbb{R}^o$ | Output of network for input $\boldsymbol{x}_i$ |
| $\mathcal{O}, o, \Omega$ and $\Theta$ | - | Standard Bachmann–Landau order notation |
| $\mathbb{P}(A)$ | - | Probability of event $A$ |

# B Proof of upper bound of the Perturbation Stability in lazy training regime for deep neural network

We present the details of our results from Section 4.1 in this section. Firstly, we introduce some lemmas in Appendix B.1 to facilitate the proof of theorems. Then, in Appendix B.2 we provide the proof of Theorem 1.

## B.1 Relevant Lemmas

**Lemma 1.** *Let $\boldsymbol{w} \sim \mathcal{N}(\boldsymbol{0}, \sigma^2 \mathbb{I}_n)$. Then, for two fixed non-zero vectors $\boldsymbol{h}_1 \in \mathbb{R}^n$ and $\boldsymbol{h}_2 \in \mathbb{R}^n$, define two random variables $X = (\boldsymbol{w}^\top \boldsymbol{h}_1 \mathbb{1}\{\boldsymbol{w}^\top \boldsymbol{h}_2 \geq 0\})^2$ and $Y = s(\boldsymbol{w}^\top \boldsymbol{h}_1)^2$, where $s \sim \mathrm{Ber}(1, 1/2)$ follows a Bernoulli distribution with $1$ trial and $\frac{1}{2}$ success rate. Then $X$ and $Y$ have the same distribution, denoted as $X \stackrel{d}{=} Y$.*

*Proof.* Firstly, we derive the cumulative distribution function (CDF) of $X$. Obviously, $X$ is non-negative and $\boldsymbol{w}^\top \boldsymbol{h}_1 \sim \mathcal{N}(\boldsymbol{0}, \sigma^2 \|\boldsymbol{h}_1\|_2^2 \mathbb{I}_n)$, and then we have:

$$\mathbb{P}(X = 0) = \mathbb{P}(\boldsymbol{w}^\top \boldsymbol{h}_2 < 0) + \mathbb{P}(\boldsymbol{w}^\top \boldsymbol{h}_2 \geq 0, \boldsymbol{w}^\top \boldsymbol{h}_1 = 0),$$

which implies:

$$\mathbb{P}(\boldsymbol{w}^\top \boldsymbol{h}_2 < 0) \leq \mathbb{P}(X = 0) \leq \mathbb{P}(\boldsymbol{w}^\top \boldsymbol{h}_2 < 0) + \mathbb{P}(\boldsymbol{w}^\top \boldsymbol{h}_1 = 0) = \mathbb{P}(\boldsymbol{w}^\top \boldsymbol{h}_2 < 0),$$

leading to $\mathbb{P}(X = 0) = \mathbb{P}(\boldsymbol{w}^\top \boldsymbol{h}_2 < 0) = 1/2$.

Accordingly, for $x \geq 0$, we have:

$$\begin{aligned}
\mathbb{P}(X \leq x) &= \mathbb{P}(\boldsymbol{w}^\top \boldsymbol{h}_2 < 0) + \mathbb{P}(\boldsymbol{w}^\top \boldsymbol{h}_2 \geq 0, -\sqrt{x} \leq \boldsymbol{w}^\top \boldsymbol{h}_1 \leq \sqrt{x}) \\
&= \frac{1}{2} + \mathbb{P}(\boldsymbol{w}^\top \boldsymbol{h}_2 \geq 0, -\sqrt{x} \leq \boldsymbol{w}^\top \boldsymbol{h}_1 \leq \sqrt{x}) \\
&= \frac{1}{2} + \mathbb{P}(\boldsymbol{w}^\top \boldsymbol{h}_2 \geq 0, -\sqrt{x} \leq \boldsymbol{w}^\top \boldsymbol{h}_1 \leq 0) + \mathbb{P}(\boldsymbol{w}^\top \boldsymbol{h}_2 \geq 0, 0 \leq \boldsymbol{w}^\top \boldsymbol{h}_1 \leq \sqrt{x}) \\
&= \frac{1}{2} + \mathbb{P}(\boldsymbol{w}^\top \boldsymbol{h}_2 \leq 0, 0 \leq \boldsymbol{w}^\top \boldsymbol{h}_1 \leq \sqrt{x}) + \mathbb{P}(\boldsymbol{w}^\top \boldsymbol{h}_2 \geq 0, 0 \leq \boldsymbol{w}^\top \boldsymbol{h}_1 \leq \sqrt{x}) \\
&= \frac{1}{2} + \mathbb{P}(0 \leq \boldsymbol{w}^\top \boldsymbol{h}_1 \leq \sqrt{x}) \\
&= \frac{1}{2} + \int_0^{\sqrt{x}} \frac{1}{\sqrt{2\pi\sigma_1^2}} e^{-\frac{t^2}{2\sigma_1^2}} \, \mathrm{d}t \,,
\end{aligned}$$

where we use the symmetry of the Gaussian random variable and $\sigma_1 = \|\boldsymbol{h}_1\|_2 \sigma$.

Then $X$ admits the following cumulative distribution function:

$$F(X \leq x) = \begin{cases} 0 & \text{if } x < 0 \\ \frac{1}{2} & \text{if } x = 0 \\ \frac{1}{2} + \int_0^{\sqrt{x}} \frac{1}{\sqrt{2\pi\sigma_1^2}} e^{-\frac{t^2}{2\sigma_1^2}} \, dt & \text{if } x > 0 \,. \end{cases} \tag{5}$$

We then derive the CDF of $Y$. Obviously, $Y$ is non-negative and $\mathbb{P}(Y = 0) = 1/2$, which holds by $Y = 0$ iff $s = 0$. Accordingly, for $x \geq 0$, we have:

$$\mathbb{P}(Y \leq x) = \mathbb{P}(s = 0) + \mathbb{P}(s = 1)\mathbb{P}(-\sqrt{x} \leq \boldsymbol{w}^\top \boldsymbol{h}_1 \leq \sqrt{x}) = \frac{1}{2} + \frac{1}{2} \int_{-\sqrt{x}}^{\sqrt{x}} \frac{1}{\sqrt{2\pi\sigma_1^2}} e^{-\frac{t^2}{2\sigma_1^2}} \, \mathrm{d}t$$

$$= \frac{1}{2} + \int_0^{\sqrt{x}} \frac{1}{\sqrt{2\pi\sigma_1^2}} e^{-\frac{t^2}{2\sigma_1^2}} \, \mathrm{d}t \,.$$

Then $Y$ has the following cumulative distribution function:

$$F(Y \leq x) = \begin{cases} 0 & \text{if } x < 0 \\ \frac{1}{2} & \text{if } x = 0 \\ \frac{1}{2} + \int_0^{\sqrt{x}} \frac{1}{\sqrt{2\pi\sigma_1^2}} e^{-\frac{t^2}{2\sigma_1^2}} dt & \text{if } x > 0, \end{cases} \tag{6}$$

which implies $X \stackrel{d}{=} Y$ by comparing Eq. (5) and Eq. (6). $\qquad\square$

**Lemma 2.** *Given two fixed non-zero vectors $\boldsymbol{h}_1 \in \mathbb{R}^p$ and $\boldsymbol{h}_2 \in \mathbb{R}^p$, let $\boldsymbol{W} \in \mathbb{R}^{q \times p}$ be random matrix with i.i.d. entries $\boldsymbol{W}_{i,j} \sim \mathcal{N}(0, 2/q)$ and a vector $\boldsymbol{v} = \phi'(\boldsymbol{W}\boldsymbol{h}_2)\boldsymbol{W}\boldsymbol{h}_1 \in \mathbb{R}^q$. Then, we have $\frac{q\|\boldsymbol{v}\|_2^2}{2\|\boldsymbol{h}_1\|_2^2} \sim \chi^2(\varrho)$, where $\varrho \sim \mathrm{Ber}(q, 1/2)$.*

*Proof.* According to the definition of $\boldsymbol{v} = \phi'(\boldsymbol{W}\boldsymbol{h}_2)\boldsymbol{W}\boldsymbol{h}_1 \in \mathbb{R}^q$, we have:

$$\|\boldsymbol{v}\|_2^2 = \sum_{i=1}^q \left( \boldsymbol{D}_{i,i} \left\langle \boldsymbol{W}^{[i]}, \boldsymbol{h}_1 \right\rangle \right)^2,$$

where $\boldsymbol{D}_{i,i} = 1\left\{ \left\langle \boldsymbol{W}^{[i]}, \boldsymbol{h}_2 \right\rangle \geq 0 \right\}$, the $\boldsymbol{W}^{[i]}$ is defined in the second part of the Table 3.

Let $\varpi_i = \left\langle \boldsymbol{W}^{[i]}, \boldsymbol{h}_1 \right\rangle / \left( \sqrt{\frac{2\|\boldsymbol{h}_1\|_2^2}{q}} \right)$, then $\varpi_i \sim \mathcal{N}(0,1)$ independently. Accordingly, by Lemma 1, recall $s \sim \mathrm{Ber}(1, 1/2)$, we have:

$$\frac{q\|\boldsymbol{v}\|_2^2}{2\|\boldsymbol{h}_1\|_2^2} = \sum_{i=1}^q \left( 1\left\{ \left\langle \boldsymbol{W}^{[i]}, \boldsymbol{h}_2 \right\rangle \geq 0 \right\} \varpi_i \right)^2 \stackrel{d}{=} \sum_{i=1}^q s\varpi_i^2,$$

which implies $\frac{q\|\boldsymbol{v}\|_2^2}{2\|\boldsymbol{h}_1\|_2^2} \sim \chi^2(\varrho)$ with $\varrho \sim \mathrm{Ber}(q, 1/2)$ according to the definition of chi-square distribution.

$\qquad\square$

**Lemma 3.** *(Dynamic equivalence under different scaling) Given an L-layer neural network $\boldsymbol{f}$ defined by Eq. (2), as follows:*
$$\boldsymbol{f}(\boldsymbol{x}) = \widehat{\boldsymbol{W}}_L \phi(\widehat{\boldsymbol{W}}_{L-1} \cdots \phi(\widehat{\boldsymbol{W}}_1 \boldsymbol{x}) \cdots), \tag{7}$$
*where $[\widehat{\boldsymbol{W}}_l]_{i,j}$ satisfy the initialization in Section 3.1, i.e., $\beta := \beta_2 = \cdots = \beta_{L-1}$.*

*Scaling all weights of $\boldsymbol{f}$, then we get a new model $\widetilde{\boldsymbol{f}}$ as follows.*
$$\widetilde{\boldsymbol{f}}(\boldsymbol{x}) = \gamma^L \widetilde{\boldsymbol{W}}_L \phi(\widetilde{\boldsymbol{W}}_{L-1} \cdots \phi(\widetilde{\boldsymbol{W}}_1 \boldsymbol{x}) \cdots), \tag{8}$$
*where $[\widetilde{\boldsymbol{W}}_l]_{i,j} = [\widehat{\boldsymbol{W}}_l]_{i,j} / \gamma \quad \forall l \in [L]$.*

*Then if we choose an appropriate learning rate $\widetilde{\eta} := \frac{\eta}{\gamma^2}$, $\boldsymbol{f}$ and $\widetilde{\boldsymbol{f}}$ will have the same dynamics.*

*Proof.* According to the chain rule, we have:
$$\frac{\mathrm{d}\widetilde{\boldsymbol{f}}}{\mathrm{d}\widetilde{\boldsymbol{W}}_l} = \gamma \frac{\mathrm{d}\boldsymbol{f}}{\mathrm{d}\widehat{\boldsymbol{W}}_l} \quad \forall l \in [L].$$

If we choose learning rate $\widetilde{\eta} := \frac{\eta}{\gamma^2}$, then, we have:

$$\frac{\mathrm{d}\widetilde{\boldsymbol{W}}_l}{\mathrm{d}t} = \frac{1}{\gamma} \frac{\mathrm{d}\widehat{\boldsymbol{W}}_l}{\mathrm{d}t} \quad \forall l \in [L].$$

Consider that $\widetilde{\boldsymbol{W}}_l(0) = \frac{1}{\gamma}\widehat{\boldsymbol{W}}_l(0) \quad \forall l \in [L]$, then, we have:
$$\widetilde{\boldsymbol{W}}_l(t) = \frac{1}{\gamma}\widehat{\boldsymbol{W}}_l(t) \quad \forall l \in [L].$$

That means $\boldsymbol{f}(t) = \widetilde{\boldsymbol{f}}(t) \quad \forall t \geq 0$, which concludes the proof. $\qquad\square$

**Lemma 4.** *Given an L-layer neural network $\boldsymbol{f}$ defined by Eq. (2) trained by $\{(\boldsymbol{x}_i, \boldsymbol{y}_i)\}_{i=1}^n$, under a small perturbation $\epsilon$, we have:*

$$\mathbb{E}_{\boldsymbol{x}, \hat{\boldsymbol{x}}, \boldsymbol{W}} \left\| \nabla_{\boldsymbol{x}} \boldsymbol{f}(\boldsymbol{x})^\top (\boldsymbol{x} - \hat{\boldsymbol{x}}) - \boldsymbol{W}_L \boldsymbol{D}_{L-1} \boldsymbol{W}_{L-1} \cdots \boldsymbol{D}_1 \boldsymbol{W}_1 (\boldsymbol{x} - \hat{\boldsymbol{x}}) \right\|_2 \leq \Theta \left( \epsilon \gamma^{L-2} \sqrt{\frac{\pi L^3 m^2 \beta_1^2 \beta_L^2}{8}} e^{-m/L^3} \right),$$

(9)

*where $[\boldsymbol{W}_l]_{i,j}$ satisfy the initialization in Section 3.1, $\boldsymbol{x} \sim \mathcal{D}_X$ and $\hat{\boldsymbol{x}} \sim Unif(\mathbb{B}(\epsilon, \boldsymbol{x}))$.*

*Proof.* We set the weight of the neural network after training are $\widehat{\boldsymbol{W}}$. i.e.

$$\boldsymbol{f}(\boldsymbol{x}) = \widehat{\boldsymbol{W}}_L \phi(\widehat{\boldsymbol{W}}_{L-1} \cdots \phi(\widehat{\boldsymbol{W}}_1 \boldsymbol{x}) \cdots).$$

According to the standard chain rule and Lemma 3, we have:

$$\nabla_{\boldsymbol{x}} \boldsymbol{f}(\boldsymbol{x})^\top = \widehat{\boldsymbol{W}}_L \widehat{\boldsymbol{D}}_{L-1} \widehat{\boldsymbol{W}}_{L-1} \cdots \widehat{\boldsymbol{D}}_1 \widehat{\boldsymbol{W}}_1 = \gamma^L \widehat{\boldsymbol{W}}_L' \widehat{\boldsymbol{D}}_{L-1} \widehat{\boldsymbol{W}}_{L-1}' \cdots \widehat{\boldsymbol{D}}_1 \widehat{\boldsymbol{W}}_1',$$

where $[\widehat{\boldsymbol{W}}_l']_{i,j} = [\widehat{\boldsymbol{W}}_l]_{i,j}/\gamma \quad \forall l \in [L]$.

Assume that the perturbation matrices satisfy $\left\| \widehat{\boldsymbol{W}}_l - \boldsymbol{W}_l \right\|_2 \leq \omega$, $\forall l \in [L]$, where the parameter $\omega$ will be determined later. Then by Allen-Zhu et al. [2019, Lemma 7.4, Lemma 8.6, Lemma 8.7], we obtain that for any integer $s \in \left[ \Omega(\frac{d}{\log m}), \mathcal{O}(\frac{m}{L^3 \log m}) \right]$, for $d \leq \mathcal{O}(\frac{m}{L \log m})$, with probability at least $1 - \exp\left( -\Omega(s \log m) \right)$ over the randomness of $\{\boldsymbol{W}\}_{l=1}^L$, it holds that:

$$\left\| \widehat{\boldsymbol{W}}_L' \widehat{\boldsymbol{D}}_{L-1} \widehat{\boldsymbol{W}}_{L-1}' \cdots \widehat{\boldsymbol{D}}_1 \widehat{\boldsymbol{W}}_1' - \boldsymbol{W}_L' \boldsymbol{D}_{L-1} \boldsymbol{W}_{L-1}' \cdots \boldsymbol{D}_1 \boldsymbol{W}_1' \right\|_2 \leq \mathcal{O}\left( \sqrt{\frac{L^3 s \log m + \omega^2 L^3 m}{d}} \sqrt{\frac{dm}{2}} \frac{\beta_1 \beta_L}{\gamma^2} \right),$$

which implies that

$$\left\| \nabla_{\boldsymbol{x}} f(\boldsymbol{x})^\top - \boldsymbol{W}_L \boldsymbol{D}_{L-1} \boldsymbol{W}_{L-1} \cdots \boldsymbol{D}_1 \boldsymbol{W}_1 \right\|_2 \leq \mathcal{O}\left( \sqrt{\frac{L^3 s \log m + \omega^2 L^3 m}{d}} \sqrt{\frac{dm}{2}} \beta_1 \beta_L \gamma^{L-2} \right),$$

holds with probability at least $1 - \exp\left( -\Omega(s \log m) \right)$.

If we choose $s := \lfloor \frac{m}{L^3 \log m} + \frac{\omega^2}{\log m} \rfloor$, then, we have:

$$\left\| \nabla_{\boldsymbol{x}} f(\boldsymbol{x})^\top - \boldsymbol{W}_L \boldsymbol{D}_{L-1} \boldsymbol{W}_{L-1} \cdots \boldsymbol{D}_1 \boldsymbol{W}_1 \right\|_2 \leq \mathcal{O}\left( \sqrt{\frac{L^3 \omega^2 + m + \omega^2 L^3 m}{d}} \sqrt{\frac{dm}{2}} \beta_1 \beta_L \gamma^{L-2} \right),$$

with probability at least $1 - \exp\left( -\Omega(\frac{m}{L^3} + \omega^2) \right)$.

Let $\delta := \sqrt{\frac{L^3 \omega^2 + m + \omega^2 L^3 m}{d}} \sqrt{\frac{dm}{2}} \beta_1 \beta_L \gamma^{L-2}$, we have $\omega^2 = \frac{u \delta^2 - m}{L^3(m+1)}$, $u = \frac{2}{m \beta_1^2 \beta_L^2 \gamma^{2(L-2)}}$. We have the following probability inequality:

$$\mathbb{P}\left( \left\| \nabla_{\boldsymbol{x}} f(\boldsymbol{x})^\top - \boldsymbol{W}_L \boldsymbol{D}_{L-1} \boldsymbol{W}_{L-1} \cdots \boldsymbol{D}_1 \boldsymbol{W}_1 \right\|_2 > \delta \right) \leq \exp\left( -\frac{u \delta^2 - m}{L^3(m+1)} - \frac{m}{L^3} \right) = \exp\left( -\frac{\delta^2 u + m^2}{L^3(m+1)} \right).$$

Then by the expectation integral equality [Vershynin, 2018, Lemma 1.2.1], the expectation is:

$$
\mathbb{E}_{\boldsymbol{W}} \left\| \nabla_{\boldsymbol{x}} f(\boldsymbol{x})^\top - \boldsymbol{W}_L \boldsymbol{D}_{L-1} \boldsymbol{W}_{L-1} \cdots \boldsymbol{D}_1 \boldsymbol{W}_1 \right\|_2 = \int_0^{+\infty} \mathbb{P}\left( \left\| \nabla_{\boldsymbol{x}} f(\boldsymbol{x})^\top - \boldsymbol{W}_L \boldsymbol{D}_{L-1} \boldsymbol{W}_{L-1} \cdots \boldsymbol{D}_1 \boldsymbol{W}_1 \right\|_2 > \delta \right) \mathrm{d}\delta
$$

$$
\leq \int_0^{+\infty} \exp\left( - \frac{\delta^2 u + m^2}{L^3(m+1)} \right) \mathrm{d}\delta
$$

$$
= \sqrt{\frac{\pi L^3 (m+1)}{4u}} \exp\left( - \frac{m^2}{(m+1)L^3} \right)
$$

$$
= \Theta\left( \gamma^{L-2} \sqrt{\frac{\pi L^3 m^2 \beta_1^2 \beta_L^2}{8}} e^{-m/L^3} \right).
$$

(10)

Finally, by the definition of $\hat{\boldsymbol{x}}$, we have:

$$
\left\| \nabla_{\boldsymbol{x}} \boldsymbol{f}^\top(\boldsymbol{x})(\boldsymbol{x} - \hat{\boldsymbol{x}}) - \boldsymbol{W}_L \boldsymbol{D}_{L-1} \boldsymbol{W}_{L-1} \cdots \boldsymbol{D}_1 \boldsymbol{W}_1 (\boldsymbol{x} - \hat{\boldsymbol{x}}) \right\|_2 \leq \epsilon \left\| \nabla_{\boldsymbol{x}} f(\boldsymbol{x})^\top - \boldsymbol{W}_L \boldsymbol{D}_{L-1} \boldsymbol{W}_{L-1} \cdots \boldsymbol{D}_1 \boldsymbol{W}_1 \right\|_2.
$$

(11)

By Eq. (10) and Eq. (11) and consider expectations for $\boldsymbol{x}$ and $\boldsymbol{x}'$, we finish the proof.

$\square$

**Lemma 5.** *Given an L-layer neural network $\boldsymbol{f}$ defined by Eq. (2) trained by $\{(\boldsymbol{x}_i, \boldsymbol{y}_i)\}_{i=1}^n$, under a small $\epsilon$, expectation over $\boldsymbol{x}, \hat{\boldsymbol{x}}, \boldsymbol{W}$, we have:*

$$
\mathbb{E}_{\boldsymbol{x}, \hat{\boldsymbol{x}}, \boldsymbol{W}} \left\| \boldsymbol{W}_L \boldsymbol{D}_{L-1} \boldsymbol{W}_{L-1} \cdots \boldsymbol{D}_1 \boldsymbol{W}_1 (\boldsymbol{x} - \hat{\boldsymbol{x}}) \right\|_2^2 \leq \frac{mo\beta_1^2 \beta_L^2 \gamma^{2(L-2)}}{2} \epsilon^2,
$$

(12)

*where $[\boldsymbol{W}_l]_{i,j}$ satisfy the initialization in Section 3.1 and $\boldsymbol{x} \sim \mathcal{D}_X$, $\hat{\boldsymbol{x}} \sim Unif(\mathbb{B}(\epsilon, \boldsymbol{x}))$.*

*Proof.* Define $\boldsymbol{t}_l = \boldsymbol{D}_l \boldsymbol{W}_l \cdots \boldsymbol{D}_1 \boldsymbol{W}_1 (\boldsymbol{x} - \hat{\boldsymbol{x}})$, then:

$$
\mathbb{E}_{\boldsymbol{x}, \hat{\boldsymbol{x}}, \boldsymbol{W}} \left\| \boldsymbol{W}_L \boldsymbol{D}_{L-1} \boldsymbol{W}_{L-1} \cdots \boldsymbol{D}_1 \boldsymbol{W}_1 (\boldsymbol{x} - \hat{\boldsymbol{x}}) \right\|_2^2 = \mathbb{E}_{\boldsymbol{x}, \hat{\boldsymbol{x}}, \boldsymbol{W}} \left\| \boldsymbol{W}_L \boldsymbol{t}_{L-1} \right\|_2^2.
$$

By Lemma 2, we have $\frac{\|\boldsymbol{t}_l\|_2^2}{\beta^2 \|\boldsymbol{t}_{l-1}\|_2^2} \sim \chi^2(\varrho)$, where $\varrho \sim \mathrm{Ber}(m, 1/2), \forall l = 2, \cdots, L-1$. By the law of total expectation $\mathbb{E}[\mathbb{E}[X|Y]] = \mathbb{E}[X]$, we have

$$
\mathbb{E}_{\boldsymbol{W}} \frac{\|\boldsymbol{t}_l\|_2^2}{\|\boldsymbol{t}_{l-1}\|_2^2} = \beta^2 \mathbb{E}_\varrho \chi^2(\varrho) = \beta^2 \mathbb{E}\varrho = \frac{m\beta^2}{2} = \gamma^2, \ \forall l = 2, \cdots, L-1.
$$

Similarly, we have:

$$
\mathbb{E}_{\boldsymbol{W}} \frac{\|\boldsymbol{t}_1\|_2^2}{\|\hat{\boldsymbol{x}} - \boldsymbol{x}\|_2^2} = \frac{m\beta_1^2}{2}.
$$

By the definition of chi-square distribution, we have $\frac{\|\boldsymbol{W}_L \boldsymbol{t}_{L-1}\|_2^2}{\beta_L^2 \|\boldsymbol{t}_{L-1}\|_2^2} \sim \chi^2(o)$, which means $\mathbb{E}_{\boldsymbol{W}} \|\boldsymbol{W}_L \boldsymbol{t}_{L-1}\|_2^2 / \|\boldsymbol{t}_{L-1}\|_2^2 = o\beta_L^2$.

Then, according to the independence among $\frac{\|\boldsymbol{W}_L \boldsymbol{t}_{L-1}\|_2^2}{\|\boldsymbol{t}_{L-1}\|_2^2}$, $\frac{\|\boldsymbol{t}_1\|_2^2}{\|\hat{\boldsymbol{x}} - \boldsymbol{x}\|_2^2}$, $\|\hat{\boldsymbol{x}} - \boldsymbol{x}\|_2^2$ and $\frac{\|\boldsymbol{t}_{l+1}\|_2^2}{\|\boldsymbol{t}_l\|_2^2}$ $\forall l \in [L-2]$, we have:

$$
\mathbb{E}_{\boldsymbol{x}, \hat{\boldsymbol{x}}, \boldsymbol{W}} \left\| \boldsymbol{W}_L \boldsymbol{t}_{L-1} \right\|_2^2 = \mathbb{E}_{\boldsymbol{x}, \hat{\boldsymbol{x}}, \boldsymbol{W}} \frac{\|\boldsymbol{W}_L \boldsymbol{t}_{L-1}\|_2^2}{\|\boldsymbol{t}_{L-1}\|_2^2} \frac{\|\boldsymbol{t}_{L-1}\|_2^2}{\|\boldsymbol{t}_{L-2}\|_2^2} \cdots \frac{\|\boldsymbol{t}_1\|_2^2}{\|\hat{\boldsymbol{x}} - \boldsymbol{x}\|_2^2} \|\hat{\boldsymbol{x}} - \boldsymbol{x}\|_2^2
$$

$$
= \mathbb{E}_{\boldsymbol{W}} \frac{\|\boldsymbol{W}_L \boldsymbol{t}_{L-1}\|_2^2}{\|\boldsymbol{t}_{L-1}\|_2^2} \mathbb{E}_{\boldsymbol{W}} \frac{\|\boldsymbol{t}_{L-1}\|_2^2}{\|\boldsymbol{t}_{L-2}\|_2^2} \cdots \mathbb{E}_{\boldsymbol{W}} \frac{\|\boldsymbol{t}_1\|_2^2}{\|\hat{\boldsymbol{x}} - \boldsymbol{x}\|_2^2} \mathbb{E}_{\boldsymbol{x}, \hat{\boldsymbol{x}}} \|\hat{\boldsymbol{x}} - \boldsymbol{x}\|_2^2
$$

$$
= \frac{mo\beta_1^2 \beta_L^2 \gamma^{2(L-2)}}{2} \mathbb{E}_{\boldsymbol{x}, \hat{\boldsymbol{x}}} \|\hat{\boldsymbol{x}} - \boldsymbol{x}\|_2^2,
$$

using the definition of $\hat{\boldsymbol{x}}$ which conclude the proof.

$\square$

## B.2 Proof of Theorem 1

*Proof.* According to the triangle inequality and the Jensen's inequality, we have:

$$
\begin{aligned}
\mathscr{P}(\boldsymbol{f},\epsilon) &= \mathbb{E}_{\boldsymbol{x},\hat{\boldsymbol{x}},\boldsymbol{W}} \|\nabla_{\boldsymbol{x}}\boldsymbol{f}(\boldsymbol{x})(\boldsymbol{x}-\hat{\boldsymbol{x}})\|_2 \\
&\leq \mathbb{E}_{\boldsymbol{x},\hat{\boldsymbol{x}},\boldsymbol{W}} \|\nabla_{\boldsymbol{x}}\boldsymbol{f}(\boldsymbol{x})(\boldsymbol{x}-\hat{\boldsymbol{x}}) - \boldsymbol{W}_L\boldsymbol{D}_{L-1}\boldsymbol{W}_{L-1}\cdots\boldsymbol{D}_1\boldsymbol{W}_1(\boldsymbol{x}-\hat{\boldsymbol{x}})\|_2 \\
&\quad + \mathbb{E}_{\boldsymbol{x},\hat{\boldsymbol{x}},\boldsymbol{W}} \|\boldsymbol{W}_L\boldsymbol{D}_{L-1}\boldsymbol{W}_{L-1}\cdots\boldsymbol{D}_1\boldsymbol{W}_1(\boldsymbol{x}-\hat{\boldsymbol{x}})\|_2 \\
&\leq \mathbb{E}_{\boldsymbol{x},\hat{\boldsymbol{x}},\boldsymbol{W}} \|\nabla_{\boldsymbol{x}}\boldsymbol{f}(\boldsymbol{x})(\boldsymbol{x}-\hat{\boldsymbol{x}}) - \boldsymbol{W}_L\boldsymbol{D}_{L-1}\boldsymbol{W}_{L-1}\cdots\boldsymbol{D}_1\boldsymbol{W}_1(\boldsymbol{x}-\hat{\boldsymbol{x}})\|_2 \\
&\quad + \sqrt{\mathbb{E}_{\boldsymbol{x},\hat{\boldsymbol{x}},\boldsymbol{W}} \|\boldsymbol{W}_L\boldsymbol{D}_{L-1}\boldsymbol{W}_{L-1}\cdots\boldsymbol{D}_1\boldsymbol{W}_1(\boldsymbol{x}-\hat{\boldsymbol{x}})\|_2^2} \\
&\lesssim \epsilon\left(\sqrt{L^3 m^2 \beta_1^2 \beta_L^2 e^{-m/L^3}} + \sqrt{m o \beta_1^2 \beta_L^2}\right)\gamma^{L-2},
\end{aligned}
$$

where the last inequality utilizes the results of Lemma 4 and Lemma 5.

$\square$

## C  Proof of sufficient condition for DNNs under the non-lazy training regime

In this section, we provide the proof of Theorem 2.

*Proof.* By Assumption 2 and by following the setting of Luo et al. [2021], without loss of generality, we have that there exits a $T^\star > 0$ such that $L(\boldsymbol{W}(T^\star)) \leq \frac{1}{32n}$ and $y_1 \geq \frac{1}{2}$. Therefore, we have:

$$
\frac{1}{2n}(f_1(T^\star) - y_1)^2 \leq \frac{1}{2n}\sum_{i=1}^{n}(f_i(T^\star) - y_i)^2 \leq L(\boldsymbol{W}(T^\star)) \leq \frac{1}{32n},
$$

which means $|f_1(T^\star) - y_1| \leq \frac{1}{4}$. Accordingly, we conclude:

$$
\begin{aligned}
\frac{1}{4} \leq y_1 - \frac{1}{4} &\leq f_1(T^\star) \\
&= \frac{1}{\alpha}\boldsymbol{W}_L(T^\star)\sigma(\boldsymbol{W}_{L-1}(T^\star)\cdots\sigma(\boldsymbol{W}_1(T^\star)\boldsymbol{x}_1)) \\
&= \frac{1}{\alpha}\boldsymbol{W}_L(T^\star)\boldsymbol{D}_{1,L-1}(T^\star)\boldsymbol{W}_{L-1}(T^\star)\cdots\boldsymbol{D}_{1,1}(T^\star)\boldsymbol{W}_1(T^\star)\boldsymbol{x}_1 \\
&\leq \frac{1}{\alpha}\|\boldsymbol{W}_L(T^\star)\|_2\|\boldsymbol{D}_{1,L-1}(T^\star)\|_2\|\boldsymbol{W}_{L-1}(T^\star)\|_2\cdots\|\boldsymbol{D}_{1,1}(T^\star)\|_2\|\boldsymbol{W}_1(T^\star)\|_2\|\boldsymbol{x}_1\|_2 \\
&\leq \frac{1}{\alpha}\|\boldsymbol{W}_L(T^\star)\|_2\|\boldsymbol{W}_{L-1}(T^\star)\|_2\cdots\|\boldsymbol{W}_1(T^\star)\|_2,
\end{aligned}
\tag{13}
$$

where the last inequality uses Assumption 1 and 1-Lipschitz of ReLU.

According to Du et al. [2018, Corollary 2.1] , we have:

$$
\frac{\mathrm{d}}{\mathrm{d}t}(\|\boldsymbol{W}_1\|_{\mathrm{F}}^2) = \frac{\mathrm{d}}{\mathrm{d}t}(\|\boldsymbol{W}_2\|_{\mathrm{F}}^2) = \cdots = \frac{\mathrm{d}}{\mathrm{d}t}(\|\boldsymbol{W}_L\|_{\mathrm{F}}^2).
$$

Then for any $l_1, l_2 \in [L]$, we have:

$$
\|\boldsymbol{W}_{l_1}(T^\star)\|_{\mathrm{F}}^2 - \|\boldsymbol{W}_{l_1}(0)\|_{\mathrm{F}}^2 = \|\boldsymbol{W}_{l_2}(T^\star)\|_{\mathrm{F}}^2 - \|\boldsymbol{W}_{l_2}(0)\|_{\mathrm{F}}^2,
$$

which implies:

$$
\begin{aligned}
\|\boldsymbol{W}_{l_1}(T^\star)\|_2 &\leq \|\boldsymbol{W}_{l_1}(T^\star)\|_{\mathrm{F}} \\
&= \sqrt{\|\boldsymbol{W}_{l_1}(T^\star)\|_{\mathrm{F}}^2} \\
&= \sqrt{\|\boldsymbol{W}_{l_2}(T^\star)\|_{\mathrm{F}}^2 - \|\boldsymbol{W}_{l_2}(0)\|_{\mathrm{F}}^2 + \|\boldsymbol{W}_{l_1}(0)\|_{\mathrm{F}}^2} \\
&\leq \sqrt{\|\boldsymbol{W}_{l_2}(T^\star)\|_{\mathrm{F}}^2 + \|\boldsymbol{W}_{l_1}(0)\|_{\mathrm{F}}^2} \\
&\leq \|\boldsymbol{W}_{l_2}(T^\star)\|_{\mathrm{F}} + \|\boldsymbol{W}_{l_1}(0)\|_{\mathrm{F}}.
\end{aligned}
\tag{14}
$$

According to Luo et al. [2021, Proposition 16] and the relationship between $\ell_2$ norm and Frobenius norm, i.e. $\|\cdot\|_F \leq \sqrt{r}\|\cdot\|_2$, where the $r$ is the rank of matrix, with probability at least $1 - (L - 2)\exp(-\Theta(m^2)) - \exp(-\Theta(md)) - \exp(-\Theta(m))$ over the initialization, we have $\|W_1(0)\|_F \leq \sqrt{d}\|W_1(0)\|_2 \leq \sqrt{\frac{3md^2}{2}}\beta_1$, $\|W_l(0)\|_F \leq \sqrt{m}\|W_l(0)\|_2 \leq \sqrt{\frac{3m^3}{2}}\beta_l$, $\forall l \in [L-1]$ and $\|W_L(0)\|_F = \|W_L(0)\|_2 \leq \sqrt{\frac{3m}{2}}\beta_L$.

If we combine Eqs. (13) and (14), for any $l^\star \in [L]$, with probability at least $1 - (L - 2)\exp(-\Theta(m^2)) - \exp(-\Theta(md)) - \exp(-\Theta(m))$ over the initialization, we have:

$$
\begin{aligned}
\frac{1}{4} &\leq \frac{1}{\alpha}\|W_L(T^\star)\|_2\|W_{L-1}(T^\star)\|_2 \cdots \|W_1(T^\star)\|_2 \\
&= \frac{1}{\alpha}\prod_{l=1}^{L}\left(\|W_{l^\star}(T^\star)\|_F + \|W_l(0)\|_F\right) \\
&\leq \frac{1}{\alpha}\left(\|W_{l^\star}(T^\star)\|_F + \frac{1}{L}\sum_{l=1}^{L}\left(\|W_l(0)\|_F\right)\right)^L \\
&\leq \frac{1}{\alpha}\left(\|W_{l^\star}(T^\star)\|_F + \sqrt{\frac{3m^3}{2L^2}}\sum_{l=1}^{L}\beta_l\right)^L.
\end{aligned}
$$

Then with probability at least $1 - (L - 2)\exp(-\Theta(m^2)) - \exp(-\Theta(md)) - \exp(-\Theta(m))$ over the initialization, we have:

$$
\|W_{l^\star}(T^\star)\|_F \geq \left(\frac{\alpha}{4}\right)^{1/L} - \sqrt{\frac{3m^3}{2L^2}}\sum_{l=1}^{L}\beta_l. \tag{15}
$$

Therefore, with probability at least $1 - (L - 2)\exp(-\Theta(m^2)) - \exp(-\Theta(md)) - \exp(-\Theta(m))$ over the initialization, we have:

$$
\begin{aligned}
\sup_{t \in [0,+\infty)}\frac{\|W_l(t) - W_l(0)\|_F}{\|W_l(0)\|_F} &\geq \frac{\|W_l(T^\star) - W_l(0)\|_F}{\|W_l(0)\|_F} \\
&\geq \frac{\|W_l(T^\star)\|_F}{\|W_l(0)\|_F} - 1 \\
&\geq \frac{\left(\frac{\alpha}{4}\right)^{1/L} - \sqrt{\frac{3m^3}{2L^2}}\sum_{i=1}^{L}\beta_i}{\sqrt{\frac{3m^3}{2}}\beta_l} - 1 \\
&\geq \frac{\left(\frac{\alpha}{4}\right)^{1/L} - \sqrt{\frac{3m^3}{2L^2}}\sum_{i=1}^{L}\beta_i}{\sqrt{\frac{3m^3}{2}}\sum_{i=1}^{L}\beta_i} - 1 \\
&= \frac{\left(\frac{\alpha}{4}\right)^{1/L}}{\sqrt{\frac{3m^3}{2}}\sum_{i=1}^{L}\beta_i} - \frac{1}{L} - 1,
\end{aligned}
$$

where the second inequality uses triangle inequality and third inequality uses Eq. (15).

If $\alpha \gg (m^{3/2}\sum_{i=1}^{L}\beta_i)^L$, then with probability at least $1 - (L-2)\exp(-\Theta(m^2)) - \exp(-\Theta(md)) - \exp(-\Theta(m))$ over the initialization, we have:

$$
\sup_{t \in [0,+\infty)}\frac{\|W_l(t) - W_l(0)\|_F}{\|W_l(0)\|_F} \gg 1.
$$

$\square$

# D Proof of the perturbation stability in non-lazy training regime for two-layer networks

Without loss of generality, we consider two-layer neural networks with a scalar output without bias.

$$f(\boldsymbol{x}) = \frac{1}{\alpha} \sum_{r=1}^{m} a_r \sigma(\boldsymbol{w}_r^\top \boldsymbol{x}) , \qquad (16)$$

where $\boldsymbol{x} \in \mathbb{R}^d$, $f(\boldsymbol{x}) \in \mathbb{R}$, $\alpha$ is the scaling factor. The parameters are initialized by $a_r(0) \sim \mathcal{N}(0, \beta_2^2)$, $\boldsymbol{w}_r(0) \sim \mathcal{N}(0, \beta_1^2 \boldsymbol{I}_d)$. Our result can be extended with slight modification to the multiple-output case with bias setting.

Our proof requires some additional notation, which we establish below:

$$\boldsymbol{H}_{ij}^\infty = \frac{m}{\alpha^2} \mathbb{E}_{\boldsymbol{w} \sim \mathcal{N}(0, \beta_1^2 \boldsymbol{I}_d), \boldsymbol{a} \sim \mathcal{N}(0, \beta_2^2)} a_r^2 \boldsymbol{x}_i^\top \boldsymbol{x}_j \mathbb{1} \left\{ \boldsymbol{w}_r^\top \boldsymbol{x}_i \geq 0, \boldsymbol{w}_r^\top \boldsymbol{x}_j \geq 0 \right\} ,$$

$$\widetilde{\boldsymbol{H}}_{i,j}(t) = \frac{1}{\alpha^2} \sum_{r=1}^{m} a_r^2(t) \mathbb{E}_{\boldsymbol{w} \sim \mathcal{N}(0, \beta_1^2 \boldsymbol{I}_d)} \boldsymbol{x}_i^\top \boldsymbol{x}_j \mathbb{1} \left\{ \boldsymbol{w}_r^\top \boldsymbol{x}_i \geq 0, \boldsymbol{w}_r^\top \boldsymbol{x}_j \geq 0 \right\} ,$$

$$\boldsymbol{H}_{i,j}(t) = \frac{1}{\alpha^2} \sum_{r=1}^{m} a_r(t)^2 \boldsymbol{x}_i^\top \boldsymbol{x}_j \mathbb{1} \left\{ \boldsymbol{w}_r(t)^\top \boldsymbol{x}_i \geq 0, \boldsymbol{w}_r(t)^\top \boldsymbol{x}_j \geq 0 \right\} ,$$

$$\widehat{\boldsymbol{H}}_{i,j} = \frac{1}{\alpha^2} \sum_{r=1}^{m} a_r(t)^2 \boldsymbol{x}_i^\top \boldsymbol{x}_j \mathbb{1} \left\{ \boldsymbol{w}_r(0)^\top \boldsymbol{x}_i \geq 0, \boldsymbol{w}_r(0)^\top \boldsymbol{x}_j \geq 0 \right\} ,$$

$$\boldsymbol{G}_{i,j}(t) = \frac{1}{\alpha^2} \sigma(\boldsymbol{w}_r(t)^\top \boldsymbol{x}_i) \sigma(\boldsymbol{w}_r(t)^\top \boldsymbol{x}_j) .$$

The minimum eigenvalue of $\boldsymbol{H}_{ij}^\infty$ is denoted as $\lambda_0$ and is assumed to be strictly greater than 0, i.e.

$$\lambda_0 = \lambda_{\min}(\boldsymbol{H}^\infty) > 0 .$$

**Remark:** This assumption follows Du et al. [2019b] but can be proved by Nguyen et al. [2021] under the NTK initialization. Moreover, Chen and Xu [2021], Geifman et al. [2020], Bietti and Mairal [2019] discuss this assumption in different settings.

The following two symbols are used to measure the weight changes during training:

$$R_a := \frac{\alpha}{n} \sqrt{\frac{\lambda_0}{8nm}} - \sqrt{\frac{2}{\pi}} \beta_2, \quad \text{and} \quad R_w := \frac{\alpha^2 \lambda_0 \sqrt{2\pi} \beta_1}{32 n^3 m (R_a(R_a + \sqrt{8/\pi} \beta_2) + \beta_2^2)} . \qquad (17)$$

The last two symbols are used to characterize the early stages of neural network training:

$$t_1^\star = -\frac{2}{\lambda_0} \log \left( 1 - \frac{R_w \lambda_0 \alpha}{2\sqrt{n}(\sqrt{n}\beta_2 + R_a) \|\boldsymbol{y} - \boldsymbol{f}(0)\|_2} \right) ,$$

$$t_2^\star = -\frac{2}{\lambda_0} \log \left( 1 - \frac{R_a \lambda_0 \alpha}{2\sqrt{n}(3\beta_1 \sqrt{\log(mn^2)} + R_w) \|\boldsymbol{y} - \boldsymbol{f}(0)\|_2} \right) .$$

Then we present the details of our results on Section 4.3 in this section. Firstly, we introduce some lemmas in Appendix D.1 to facilitate the proof of theorems. Then in Appendix D.2 we provide the proof of Theorem 3.

## D.1 Relevant Lemmas

**Lemma 6.** *[Du et al., 2019b, Appendix A.1] Given a two-layer neural network $f$ defined by Eq. (16) and trained by $\{\boldsymbol{x}_i, y_i\}_{i=1}^{n}$ using gradient descent with the quadratic loss, let $\boldsymbol{y} = (y_1, \ldots, y_n) \in \mathbb{R}^n$ be the label vector and $\boldsymbol{f}(t) = (f_1(t), \ldots, f_n(t)) \in \mathbb{R}^n$ be the output vector at time t, then, we have:*

$$\frac{\mathrm{d}\boldsymbol{f}(t)}{\mathrm{d}t} = (\boldsymbol{H}(t) + \boldsymbol{G}(t))(\boldsymbol{y} - \boldsymbol{f}(t)). \tag{18}$$

*Proof.* Our proof here just re-organizes Du et al. [2019b, Appendix A.1]. For making our manuscript self-contained, we provide a formal proof here.

We want to minimize the quadratic loss:

$$L(\boldsymbol{W}, \boldsymbol{a}) = \sum_{i=1}^{n} \frac{1}{2}[f(\boldsymbol{W}, \boldsymbol{a}, \boldsymbol{x}_i) - y_i]^2.$$

Using the gradient descent algorithm, the formula for update the weights is:

$$\boldsymbol{W}(t+1) = \boldsymbol{W}(t) - \eta \frac{\partial L(\boldsymbol{W}(t), \boldsymbol{a}(t))}{\partial \boldsymbol{W}(t)},$$

$$\boldsymbol{a}(t+1) = \boldsymbol{a}(t) - \eta \frac{\partial L(\boldsymbol{W}(t), \boldsymbol{a}(t))}{\partial \boldsymbol{a}(t)}.$$

According to the standard chain rule, we have:

$$\frac{\partial L(\boldsymbol{W}(t), \boldsymbol{a}(t))}{\partial \boldsymbol{W}(t)} = \frac{1}{\alpha} \sum_{i=1}^{n} [f(\boldsymbol{W}(t), \boldsymbol{a}(t), \boldsymbol{x}_i) - y_i] a_r(t) \mathbb{1}\left\{\boldsymbol{w}_r^\top(t)\boldsymbol{x}_i \geq 0\right\} \boldsymbol{x}_i,$$

$$\frac{\partial L(\boldsymbol{W}(t), \boldsymbol{a}(t))}{\partial \boldsymbol{a}(t)} = \frac{1}{\alpha} \sum_{i=1}^{n} [f(\boldsymbol{W}(t), \boldsymbol{a}(t), \boldsymbol{x}_i) - y_i] \sigma(\boldsymbol{w}_r^\top(t)\boldsymbol{x}_i).$$

Then, we have:

$$\begin{aligned}
\frac{\mathrm{d}f_i(t)}{\mathrm{d}t} &= \sum_{r=1}^{m} \left\langle \frac{\partial f_i(t)}{\partial \boldsymbol{w}_r(t)}, \frac{\partial \boldsymbol{w}_r(t)}{\partial t} \right\rangle + \sum_{r=1}^{m} \frac{\mathrm{d}f_i(t)}{\mathrm{d}a_r(t)} \frac{\mathrm{d}a_r(t)}{\mathrm{d}t} \\
&= \sum_{i=1}^{n} [y_i - f_i(t)][\boldsymbol{H}_{ij}(t) + \boldsymbol{G}_{ij}(t)].
\end{aligned}$$

Written in vector form, we have:

$$\frac{\mathrm{d}\boldsymbol{f}(t)}{\mathrm{d}t} = (\boldsymbol{H}(t) + \boldsymbol{G}(t))(\boldsymbol{y} - \boldsymbol{f}(t)).$$

$\square$

**Lemma 7.** *If $\alpha \geq \frac{16n\beta_2\sqrt{\log(2n^3)}}{\lambda_0}$, with probability at least $1 - \frac{1}{n}$, we have:*

$$\|\boldsymbol{H}(0) - \boldsymbol{H}^\infty\|_2 \leq \frac{\lambda_0}{4}, \quad \text{and} \quad \lambda_{\min}(\boldsymbol{H}(0)) \geq \frac{3}{4}\lambda_0,$$

**Remark:** This lemma is a modified version of Du et al. [2019b, Lemma 3.1], which differs in the initialization of $\boldsymbol{a}$ from $\mathrm{Unif}(\{-1, +1\})$ to Gaussian initialization. This makes our analysis relatively intractable due to their analysis based on $a_i^2 = 1, \forall i \in [m]$.

*Proof.* Firstly, for a fixed pair $(i,j)$, $\boldsymbol{H}_{ij}^{\infty}$ is an average of $\widetilde{\boldsymbol{H}}_{i,j}$ with respect to $a_r$. By Bernstein's inequality [Vershynin, 2018, Chapter 2], with probability at least $1 - \delta$, we have:

$$\left|\boldsymbol{H}_{ij}^{\infty} - \widetilde{\boldsymbol{H}}_{i,j}\right| \leq \frac{2\beta_2\sqrt{\log(\frac{1}{\delta})}}{\alpha} \, .$$

Then, for fixed pair $(i,j)$, $\widetilde{\boldsymbol{H}}_{i,j}$ is an average of $\boldsymbol{H}_{ij}(0)$ with respect to $\boldsymbol{w}_r$. By Hoeffding's inequality [Vershynin, 2018, Chapter 2], with probability at least $1 - \delta'$, we have:

$$\left|\boldsymbol{H}_{ij}(0) - \widetilde{\boldsymbol{H}}_{i,j}\right| \leq \frac{2\beta_2\sqrt{\log(\frac{1}{\delta'})}}{\alpha} \, .$$

Choose $\delta := \delta' := \frac{1}{2n^3}$, we have with probability at least $1 - \frac{1}{n^3}$, for fixed pair $(i,j)$:

$$\left|\boldsymbol{H}_{ij}(0) - \boldsymbol{H}_{ij}^{\infty}\right| \leq \frac{4\beta_2\sqrt{\log(2n^3)}}{\alpha} \, .$$

Consider the union bound over $(i,j)$ pairs, with probability at least $1 - \frac{1}{n}$, we have:

$$\left|\boldsymbol{H}_{ij}(0) - \boldsymbol{H}_{ij}^{\infty}\right| \leq \frac{4\beta_2\sqrt{\log(2n^3)}}{\alpha} \, .$$

Thus, we have:

$$\|\boldsymbol{H}(0) - \boldsymbol{H}^{\infty}\|_2^2 \leq \|\boldsymbol{H}(0) - \boldsymbol{H}^{\infty}\|_{\mathrm{F}}^2 \leq \sum_{i,j}\left|\boldsymbol{H}_{ij}(0) - \boldsymbol{H}_{ij}^{\infty}\right|^2 \leq \frac{16n^2\beta_2^2\log(2n^3)}{\alpha^2} \, .$$

when $\alpha \geq \frac{16n\beta_2\sqrt{\log(2n^3)}}{\lambda_0}$, we have the desired result. $\qquad\square$

**Lemma 8.** *With probability at least $1 - \frac{2}{n}$ over initialization, if a set of weight vectors $\{\boldsymbol{w}_r\}_{r=1}^m$ and the output weight $\{a_r\}_{r=1}^m$ satisfy for all $r \in [m]$, $\|\boldsymbol{w}_r(t) - \boldsymbol{w}_r(0)\|_2 \leq R_w$ and $|a_r(t) - a_r(0)| \leq R_a$, then, we have:*

$$\|\boldsymbol{H}(t) - \boldsymbol{H}(0)\|_2 \leq \frac{\lambda_0}{4}, \quad and \quad \lambda_{\min}(\boldsymbol{H}(t)) \geq \frac{\lambda_0}{2} \, .$$

*Proof.* Firstly, we can derive that:

$$\widehat{\boldsymbol{H}}_{i,j}(t) - \boldsymbol{H}_{i,j}(0) = \frac{1}{\alpha^2}\sum_{r=1}^m (a_r(t)^2 - a_r(0)^2)\boldsymbol{x}_i^{\top}\boldsymbol{x}_j \mathbb{1}\left\{\boldsymbol{w}_r(0)^{\top}\boldsymbol{x}_i \geq 0, \boldsymbol{w}_r(0)^{\top}\boldsymbol{x}_j \geq 0\right\} ,$$

$$\begin{aligned}
\boldsymbol{H}_{i,j}(t) - \widehat{\boldsymbol{H}}_{i,j}(t) &= \frac{1}{\alpha^2}\sum_{r=1}^m a_r(t)^2 \boldsymbol{x}_i^{\top}\boldsymbol{x}_j \mathbb{1}\left\{\boldsymbol{w}_r(t)^{\top}\boldsymbol{x}_i \geq 0, \boldsymbol{w}_r(t)^{\top}\boldsymbol{x}_j \geq 0\right\} \\
&\quad - \frac{1}{\alpha^2}\sum_{r=1}^m a_r(t)^2 \boldsymbol{x}_i^{\top}\boldsymbol{x}_j \mathbb{1}\left\{\boldsymbol{w}_r(0)^{\top}\boldsymbol{x}_i \geq 0, \boldsymbol{w}_r(0)^{\top}\boldsymbol{x}_j \geq 0\right\} \, .
\end{aligned}$$

Then we can compute the expectation of $\left|\widehat{\boldsymbol{H}}_{i,j}(t) - \boldsymbol{H}_{i,j}(0)\right|$:

$$\mathbb{E}\left|\widehat{\boldsymbol{H}}_{i,j}(t) - \boldsymbol{H}_{i,j}(0)\right| = \mathbb{E}\left|\frac{1}{\alpha^2}\sum_{r=1}^{m}(a_r(t)^2 - a_r(0)^2)\boldsymbol{x}_i^\top\boldsymbol{x}_j1\left\{\boldsymbol{w}_r(0)^\top\boldsymbol{x}_i \geq 0, \boldsymbol{w}_r(0)^\top\boldsymbol{x}_j \geq 0\right\}\right|$$

$$\leq \frac{m}{\alpha^2}\mathbb{E}\left|a_r(t)^2 - a_r(0)^2\right|$$

$$= \frac{m}{\alpha^2}\mathbb{E}\left|(a_r(t) - a_r(0))(a_r(t) + a_r(0))\right|$$

$$\leq \frac{mR_a}{\alpha^2}\mathbb{E}\left|a_r(t) + a_r(0)\right|$$

$$\leq \frac{mR_a}{\alpha^2}(R_a + 2\mathbb{E}\left|a_r(0)\right|)$$

$$\leq \frac{m(R_a + \mathbb{E}\left|a_r(0)\right|)^2}{\alpha^2}$$

$$= \frac{m(R_a + \sqrt{\frac{2}{\pi}}\beta_2)^2}{\alpha^2}\,.$$

(19)

Then we define the event:

$$A_{i,r} = \left\{\exists : \|\boldsymbol{w}_r(t) - \boldsymbol{w}_r(0)\| \leq R_w, 1\left\{\boldsymbol{w}_r(0)^\top\boldsymbol{x}_i \geq 0\right\} \neq 1\left\{\boldsymbol{w}_r(t)^\top\boldsymbol{x}_i \geq 0\right\}\right\}\,.$$

This event happens if and only if $\left|\boldsymbol{w}_r(0)^\top\boldsymbol{x}_i\right| < R_t$. According to this, we can get $\mathbb{P}(A_{i,r}) = \mathbb{P}_{z\sim\mathcal{N}(0,\beta_1^2)}(|z| \leq R_w) \leq \frac{2R_w}{\sqrt{2\pi}\beta_1}$, further:

$$\mathbb{E}\left|\boldsymbol{H}_{i,j}(t) - \widehat{\boldsymbol{H}}_{i,j}(t)\right| = \frac{1}{\alpha^2}\mathbb{E}\left|\sum_{r=1}^{m}a_r(t)^2\boldsymbol{x}_i^\top\boldsymbol{x}_j1\left\{\boldsymbol{w}_r(t)^\top\boldsymbol{x}_i \geq 0, \boldsymbol{w}_r(t)^\top\boldsymbol{x}_j \geq 0\right\}\right.$$

$$\left.-\sum_{r=1}^{m}a_r(t)^2\boldsymbol{x}_i^\top\boldsymbol{x}_j1\left\{\boldsymbol{w}_r(0)^\top\boldsymbol{x}_i \geq 0, \boldsymbol{w}_r(0)^\top\boldsymbol{x}_j \geq 0\right\}\right|$$

$$\leq \frac{1}{\alpha^2}\sum_{r=1}^{m}\mathbb{E}\left(a_r(t)^2\boldsymbol{x}_i^\top\boldsymbol{x}_j1\left\{A_{i,r} \cup A_{j,r}\right\}\right)$$

$$\leq \frac{1}{\alpha^2}\sum_{r=1}^{m}\mathbb{E}\left(a_r(t)^2\frac{4R_w}{\sqrt{2\pi}\beta_1}\right)$$

(20)

$$= \frac{4R_w}{\alpha^2\sqrt{2\pi}\beta_1}\sum_{r=1}^{m}\mathbb{E}(a_r(t)^2 - a_r(0)^2 + a_r(0)^2)$$

$$\leq \frac{4R_wm}{\alpha^2\sqrt{2\pi}\beta_1}\left(R_a(R_a + \sqrt{\frac{8}{\pi}}\beta_2) + \beta_2^2\right),$$

where the last inequality uses the result of Eq. (19).

From Eqs. (19) and (20), using Markov's inequality. with probability at least $1 - \frac{2}{n}$, we have:

$$\left|\widehat{\boldsymbol{H}}_{i,j}(t) - \boldsymbol{H}_{i,j}(0)\right| \leq \frac{nm(R_a + \sqrt{\frac{2}{\pi}}\beta_2)^2}{\alpha^2}\,,$$

$$\left|\boldsymbol{H}_{i,j}(t) - \widehat{\boldsymbol{H}}_{i,j}(t)\right| \leq \frac{4R_wnm}{\alpha^2\sqrt{2\pi}\beta_1}\left(R_a(R_a + \sqrt{\frac{8}{\pi}}\beta_2) + \beta_2^2\right)\,.$$

Then, we have:

$$
\begin{aligned}
\|\boldsymbol{H}(t) - \boldsymbol{H}(0)\|_2 &\leq \|\boldsymbol{H}(t) - \boldsymbol{H}(0)\|_{\mathrm{F}} \\
&\leq \sum_{(i,j)=(1,1)}^{(n,n)} |\boldsymbol{H}_{i,j}(t) - \boldsymbol{H}_{i,j}(0)| \\
&\leq \sum_{(i,j)=(1,1)}^{(n,n)} \left( \left|\widehat{\boldsymbol{H}}_{i,j}(t) - \boldsymbol{H}_{i,j}(0)\right| + \left|\boldsymbol{H}_{i,j}(t) - \widehat{\boldsymbol{H}}_{i,j}(t)\right| \right) \\
&\leq \frac{mn^3}{\alpha^2} \left( (R_a + \sqrt{\tfrac{2}{\pi}}\beta_2)^2 + \frac{4R_w}{\sqrt{2\pi}\beta_1}(R_a(R_a + \sqrt{\tfrac{8}{\pi}}\beta_2) + \beta_2^2) \right).
\end{aligned}
$$

Then, by Eq. (17), we have:

$$
\|\boldsymbol{H}(t) - \boldsymbol{H}(0)\|_2 \leq \frac{\lambda_0}{4},
$$

which implies:

$$
\lambda_{\min}(\boldsymbol{H}(t)) \leq \lambda_{\min}(\boldsymbol{H}(0)) - \frac{\lambda_0}{4} \leq \frac{\lambda_0}{2}.
$$

$\square$

**Lemma 9.** *Suppose that for $0 \leq s \leq t$, $\lambda_{\min}(\boldsymbol{H}(s)) \geq \frac{\lambda_0}{2}$ and $|a_r(s) - a_r(0)| \leq R_a$. Then with probability at least $1 - n\exp(-n/2)$ over initialization, we have $\|\boldsymbol{w}_r(t) - \boldsymbol{w}_r(0)\|_2 \leq R_w$ for all $r \in [m]$ and the $t \leq t_1^\star$.*

*Proof.* By Lemma 6, we have $\frac{\mathrm{d}\boldsymbol{f}(t)}{\mathrm{d}t} = (\boldsymbol{H}(t) + \boldsymbol{G}(t))(\boldsymbol{y} - \boldsymbol{f}(t))$. Then we can calculate the dynamics of risk function:

$$
\begin{aligned}
\frac{\mathrm{d}}{\mathrm{d}t} \|\boldsymbol{y} - \boldsymbol{f}(t)\|_2^2 &= -2(\boldsymbol{y} - \boldsymbol{f}(t))^\top (\boldsymbol{H}(t) + \boldsymbol{G}(t))(\boldsymbol{y} - \boldsymbol{f}(t)) \\
&\leq -2(\boldsymbol{y} - \boldsymbol{f}(t))^\top (\boldsymbol{H}(t))(\boldsymbol{y} - \boldsymbol{f}(t)) \\
&\leq -\lambda_0 \|\boldsymbol{y} - \boldsymbol{f}(t)\|_2^2,
\end{aligned}
$$

in the first inequality we use that the $\boldsymbol{G}(t)$ is Gram matrix thus it is positive. Then, we have $\frac{\mathrm{d}}{\mathrm{d}t}\left(e^{\lambda_0 t} \|\boldsymbol{y} - \boldsymbol{f}(t)\|_2^2\right) \leq 0$, then $e^{\lambda_0 t} \|\boldsymbol{y} - \boldsymbol{f}(t)\|_2^2$ is a decreasing function with respect to $t$. Thus, we can bound the risk:

$$
\|\boldsymbol{y} - \boldsymbol{f}(t)\|_2^2 \leq e^{-\lambda_0 t} \|\boldsymbol{y} - \boldsymbol{f}(0)\|_2^2. \tag{21}
$$

Then we bound the gradient of $\boldsymbol{w}_r$. For $0 \leq s \leq t$, With probability at least $1 - n\exp(-n/2)$, we have:

$$
\begin{aligned}
\left\|\frac{\mathrm{d}}{\mathrm{d}s}\boldsymbol{w}_r(s)\right\|_2 &= \left\|\frac{1}{\alpha}\sum_{i=1}^{n}[f(\boldsymbol{W}(s), \boldsymbol{a}(s), \boldsymbol{x}_i) - y_i]a_r(s)\mathbf{1}\left\{\boldsymbol{w}_r^\top(s)\boldsymbol{x}_i \geq 0\right\}\boldsymbol{x}_i\right\|_2 \\
&\leq \frac{1}{\alpha}\sum_{i=1}^{n} |f(\boldsymbol{W}(s), \boldsymbol{a}(s), \boldsymbol{x}_i) - y_i|\,|a_r(0) + R_a| \\
&\leq \frac{\sqrt{n}}{\alpha} \|\boldsymbol{y} - \boldsymbol{f}(s)\|_2 (\sqrt{n}\beta_2 + R_a) \\
&\leq \frac{\sqrt{n}}{\alpha} (\sqrt{n}\beta_2 + R_a)e^{-\lambda_0 s/2} \|\boldsymbol{y} - \boldsymbol{f}(0)\|_2,
\end{aligned}
$$

where the second inequality is because of $a_r(0) \sim \mathcal{N}(0, \beta_2^2)$, then with probability at least $1 - \exp(-n/2)$, we have $a_r(0) \leq \sqrt{n}\beta_2$. Then, we have:

$$\|\boldsymbol{w}_r(t) - \boldsymbol{w}_r(0)\|_2 \leq \int_0^t \left\| \frac{\mathrm{d}}{\mathrm{d}s} \boldsymbol{w}_r(s) \right\|_2 \mathrm{d}s \leq \frac{2\sqrt{n}}{\lambda_0 \alpha} (\sqrt{n}\beta_2 + R_a) \|\boldsymbol{y} - \boldsymbol{f}(0)\|_2 (1 - \exp(-\frac{\lambda_0 t}{2})). \tag{22}$$

If we account for $t$, then we conclude the proof. $\qquad\square$

**Lemma 10.** *Suppose that for* $0 \leq s \leq t$, $\lambda_{\min}(\boldsymbol{H}(s)) \geq \frac{\lambda_0}{2}$ *and* $\|\boldsymbol{w}_r(s) - \boldsymbol{w}_r(0)\|_2 \leq R_w$. *Then with probability at least* $1 - \frac{1}{n}$ *over initialization, we have* $|a_r(t) - a_r(0)| \leq R_a$ *for all* $r \in [m]$ *and the* $t \leq t_2^\star$.

*Proof.* Note for any $i \in [n]$ and $r \in [m]$, $\boldsymbol{w}_r^\top(0)\boldsymbol{x}_i \sim \mathcal{N}(0, \beta_1^2)$. Therefore applying Gaussian tail bound and union bound, we have with probability at least $1 - \frac{1}{n}$, for all $i \in [n]$ and $r \in [m]$, $\left|\boldsymbol{w}_r^\top(0)\boldsymbol{x}_i\right| \leq 3\beta_1\sqrt{\log(mn^2)}$, That means for $0 \leq s \leq t$, With probability at least $1 - \frac{1}{n}$, we have:

$$\begin{aligned}
\left| \frac{\mathrm{d}}{\mathrm{d}s} a_r(s) \right| &= \left| \frac{1}{\alpha} \sum_{i=1}^n [f(\boldsymbol{W}(t), \boldsymbol{a}(t), \boldsymbol{x}_i) - y_i] \sigma(\boldsymbol{w}_r^\top(t)\boldsymbol{x}_i) \right| \\
&\leq \frac{\sqrt{n}}{\alpha} \|\boldsymbol{y} - \boldsymbol{f}(s)\|_2 \left( \left|\boldsymbol{w}_r^\top(0)\boldsymbol{x}_i\right| + R_w \right) \\
&\leq \frac{\sqrt{n}}{\alpha} e^{-\lambda_0 s/2} \|\boldsymbol{y} - \boldsymbol{f}(0)\|_2 \left( 3\beta_1\sqrt{\log(mn^2)} + R_w \right).
\end{aligned}$$

Then, we have:

$$|a_r(t) - a_r(0)|_2 \leq \int_0^t \left| \frac{\mathrm{d}}{\mathrm{d}s} a_r(s) \right| \mathrm{d}s \leq \frac{2\sqrt{n}}{\lambda_0 \alpha} \left( 3\beta_1\sqrt{\log(mn^2)} + R_w \right) \|\boldsymbol{y} - \boldsymbol{f}(0)\|_2 (1 - \exp(-\frac{\lambda_0 t}{2})). \tag{23}$$

Bring in $t$, then finish the proof.

$\qquad\square$

**Lemma 11.** *Suppose* $0 \leq t \leq \min(t_1^\star, t_2^\star)$. *Then with probability at least* $1 - n\exp(-n/2) - \frac{3}{n}$ *over initialization, we have:* $\lambda_{\min}(\boldsymbol{H}(t)) \geq \frac{\lambda_0}{2}$,

$$|a_r(t) - a_r(0)| \leq \frac{2\sqrt{n}}{\lambda_0 \alpha} \left( 3\beta_1\sqrt{\log(mn^2)} + R_w \right) \|\boldsymbol{y} - \boldsymbol{f}(0)\|_2 (1 - \exp(-\frac{\lambda_0 t}{2})) := R_a^\star(t),$$

$$\|\boldsymbol{w}_r(t) - \boldsymbol{w}_r(0)\|_2 \leq \frac{2\sqrt{n}}{\lambda_0 \alpha} (\sqrt{n}\beta_2 + R_a) \|\boldsymbol{y} - \boldsymbol{f}(0)\|_2 (1 - \exp(-\frac{\lambda_0 t}{2})) := R_w^\star(t),$$

*for all* $r \in [m]$.

*Proof.* When $t = 0$, $\lambda_{\min}(\boldsymbol{H}(s)) \geq \frac{3}{4}\lambda_0$, $|a_r(t) - a_r(0)| = 0 < R_a$ and $\|\boldsymbol{w}_r(t) - \boldsymbol{w}_r(0)\|_2 = 0 < R_w$. Using induction, combine Lemma 8, Lemma 9 and Lemma 10, we have the result. $\qquad\square$

### D.2 Proof of Theorem 3

*Proof.* We can compute the gradient of the network that:

$$\nabla_{\boldsymbol{x}} f_t(\boldsymbol{x}) = \frac{1}{\alpha} \sum_{r=1}^m a_r(t) \mathbb{1}\left\{\boldsymbol{w}_r^\top(t)\boldsymbol{x} \geq 0\right\} \boldsymbol{w}_r^\top(t).$$

$\qquad\square$

Then we can derive that:

$$
\begin{aligned}
\mathscr{P}(f_t, \epsilon) &= \mathbb{E}_{\boldsymbol{x}, \hat{\boldsymbol{x}}} \left| \frac{1}{\alpha} \sum_{r=1}^{m} a_r(t) \mathbf{1}\left\{ \boldsymbol{w}_r^\top(t)\boldsymbol{x} \geq 0 \right\} \boldsymbol{w}_r^\top(t)(\boldsymbol{x} - \hat{\boldsymbol{x}}) \right| \\
&\leq \frac{1}{\alpha} \mathbb{E}_{\boldsymbol{x}, \hat{\boldsymbol{x}}} \sum_{r=1}^{m} \left| a_r(t) \boldsymbol{w}_r^\top(t)(\boldsymbol{x} - \hat{\boldsymbol{x}}) \right| \\
&\leq \frac{1}{\alpha} \mathbb{E}_{\boldsymbol{x}, \hat{\boldsymbol{x}}} \sum_{r=1}^{m} \left| a_r(t) \right| \left\| \boldsymbol{w}_r(t) \right\|_2 \left\| \boldsymbol{x} - \hat{\boldsymbol{x}} \right\|_2 \\
&\leq \frac{\epsilon}{\alpha} \sum_{r=1}^{m} \left| a_r(t) \right| \left\| \boldsymbol{w}_r(t) \right\|_2 .
\end{aligned}
\tag{24}
$$

Then by Lemma 11, we have:

$$
\left| a_r(t) \right| \leq \left| a_r(t) - a_r(0) \right| + \left| a_r(0) \right| \leq R_a^\star(t) + \left| a_r(0) \right| .
$$

$$
\left\| \boldsymbol{w}_r(t) \right\|_2 \leq \left\| \boldsymbol{w}_r(t) - \boldsymbol{w}_r(0) \right\|_2 + \left\| \boldsymbol{w}_r(0) \right\|_2 \leq R_w^\star(t) + \left\| \boldsymbol{w}_r(0) \right\|_2 .
$$

From Eq. (19), we have $\mathbb{E} \left| a_r(0) \right| = \sqrt{\frac{2}{\pi}} \beta_2$. That means with probability at least $1 - \frac{1}{n}$ over initialization, we have $\left| a_r(0) \right| \leq \sqrt{\frac{2}{\pi}} n \beta_2$.

By Vershynin [2018, Chapter 3], with probability at least $1 - \delta$ over initialization, we have $\left\| \boldsymbol{w}_r(0) \right\|_2 \leq 4\beta_1 \sqrt{m} + 2\beta_1 \sqrt{\log n}$.

By combining the results above with Eq. (24) and Lemma 11, with probability at least $1 - n \exp(-n/2) - \frac{3}{n}$ over initialization we obtain that:

$$
\begin{aligned}
\mathscr{P}(f_t, \epsilon) &\leq \frac{\epsilon}{\alpha} \sum_{r=1}^{m} \left| a_r(t) \right| \left\| \boldsymbol{w}_r(t) \right\|_2 \\
&\leq \frac{\epsilon m}{\alpha} \left( R_a^\star(t) + \sqrt{\frac{2}{\pi}} n \beta_2 \right) \left( R_w^\star(t) + 4\beta_1 \sqrt{m} + 2\beta_1 \sqrt{\log n} \right)
\end{aligned}
\tag{25}
$$

Suppose that $\alpha \sim 1$, $\beta_1 \sim \beta_2 \sim \beta \sim \frac{1}{m^c}$, $c \geq 1.5$, $m \gg n^2$. Then $R_a = \Theta(\frac{1}{\sqrt{n^3 m}})$, $R_w = \Theta(\frac{1}{m^c})$, $R_a^\star(t) = \Theta(\frac{\sqrt{n \log m}}{m^c})$ and $R_w^\star(t) = \Theta(\frac{1}{\sqrt{n^3 m}})$. Bring these results into Eq. (25), with probability at least $1 - n \exp(-\frac{n}{2}) - \frac{3}{n}$ over initialization, we have:

$$
\mathscr{P}(f_t, \epsilon) \leq \Theta \left( \epsilon \frac{\sqrt{n \log m} + n}{m^{c-1}} \left( \frac{1}{\sqrt{n^3 m}} + \frac{1}{m^{c-0.5}} \right) \right) .
$$

# E  Additional Experiments

A number of additional experiments are conducted in this section. Unless explicitly mentioned otherwise, the experimental setup remains similar to the one in the main paper. The following experiments are conducted below:

1. In Appendix E.1, we compare the two different training regimes, lazy training and non-lazy training.

2. In Appendix E.2, we conduct some experiments to assess early-stopping training.

3. In Appendix E.3, we extend the experiments in Section 5.3 from fully connected network to Convolutional Neural Network.

Table 4: Compare the clean accuracy for lazy training regime and non-lazy training regime of ResNet-110.

| Dataset | Lazy training | Non-lazy training |
|---------|---------------|-------------------|
| CIFAR10 | 92.89% | 92.14% |
| CIFAR100 | 71.08% | 70.55% |

4. In Appendix E.5, we conduct experiments with additional initializations under the non-lazy training regime.

5. In Appendix E.6, we extend the experiments from He and LeCun initialization to NTK initialization.

6. In Appendix E.4, we extend in Section 5.3 from fully connected network to ResNet.

### E.1 Comparison of Lazy training and Non-lazy training

In this section, we test the performance of the lazy training regime and the non-lazy training regime on the standard ResNet-110[3]. We adopt a narrow model width for computational efficiency. We choose the He initialization and the non-lazy training initialization as mentioned in Section 5.1 on CIFAR10 and CIFAR100. The results are provided in Table 4. Notice that the non-lazy training regime achieves a similar performance to the lazy training regime. This implies that the non-lazy training regime is also needed for studying practical learning tasks.

### E.2 Ablation study on early stopping (training 50 epochs)

In this section, we conduct an experiment to assess the early-stopping technique that is frequently employed in neural network training. In our case, we consider stopping after 50 epochs. The experimental results shown in Fig. 6 indicate that the loss and accuracy of the neural network remain almost unchanged from the 50th epoch to the 200th epoch under two different network settings: width = 32, depth = 4 and width = 64, depth = 8. Therefore, we train the rest of the networks for 50 epochs in this work.

### E.3 Extension of Section 5.3 to convolutional networks

We extend the experiments of Section 5.3 from fully connected networks to convolutional neural networks in Fig. 7. Compared with the fully connected network, the main difference of the convolutional neural network is that the difference between different depths is much larger than fully connected network, which is more in line with the relationship between robustness and depth under He initialization in Theorem 1.

### E.4 Additional experiments on ResNet

In this section, we extend the experiments in Section 5.3 from fully connected networks to ResNet in Fig. 8. Compared with the fully connected network, the results of ResNet show similar characteristics to our theory on fully connected networks. Specifically, the perturbation stability increases with depth, and an insignificant phase transition can also be seen for width.

### E.5 Additional experiments in non-lazy training regime

We extend the experiments of Fig. 4(b) to more initializations under non-lazy training regime (the variance of the initial weight are $\frac{1}{m^3}$ and $\frac{1}{m^4}$). Fig. 9 provides the relationship between robustness and width of neural network for these two initializations and shows that the robustness improves with the increase of the width of network which is consistent with Theorem 3. However, the difference between different initializations is not as large as our theoretical expectation, which may indicate that the bound in Theorem 3 is not tight enough.

---

[3]We use the following link for the implementation: `https://github.com/bearpaw/pytorch-classification`.

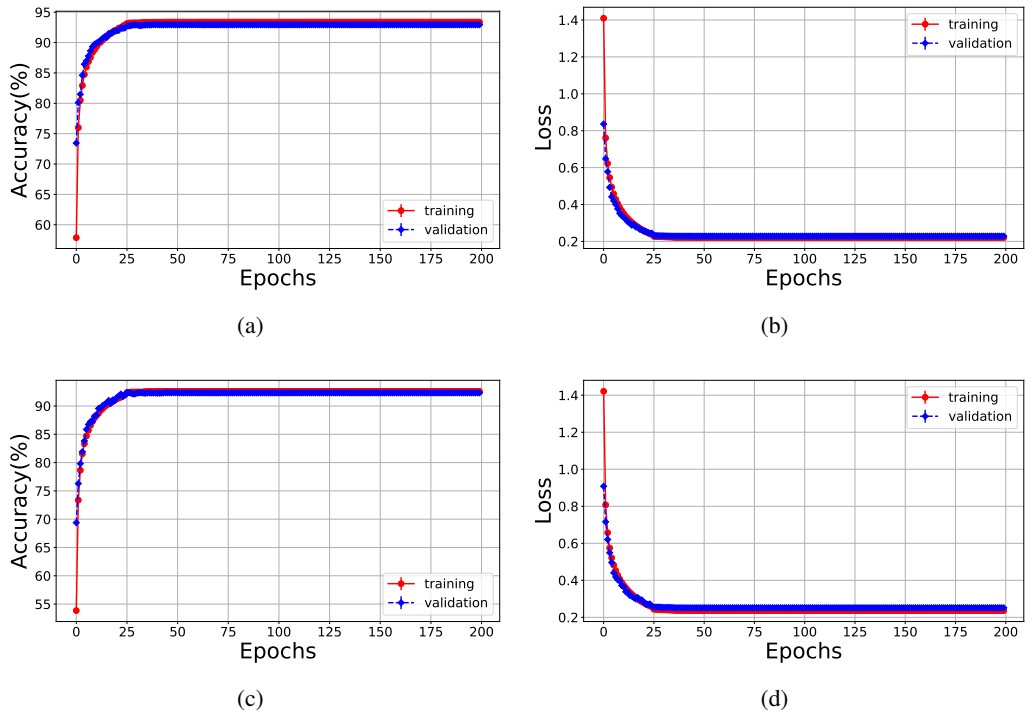

(a)                 (b)

(c)                 (d)

Figure 6: Effect of the early-stop of training on (a) the accuracy and (b) the loss of the network with width=32 and depth=4, (c) the accuracy and (d) the loss of the network with width=64 and depth=8.

### E.6 Additional experiments under NTK initialization

In this section, we extend the experiments in Fig. 5 from He and LeCun initialization to NTK initialization in Fig. 10. Our experimental results show that, NTK initialization and He initialization yield similar curves, but differ in the curve of $L = 2$. This may be because the infinite-width NTK is equivalent to the linear model, and the large finite-width network approximates the linear model. This phenomenon can be more easily detected for two-layer neural networks when compared to deeper networks.

## F Limitation and discussion

The limitation of this work is mainly manifested in that Theorem 3 is built on two-layers neural networks. Extending this results to deep neural networks beyond lazy training regime is non-trivial. Firstly, the dynamics of the deep neural network and the bounds of the gap between the initialization and the expectation of the gram matrix will become more complex. Secondly, due to the coupling relationship between different layers, the critical change radius of the weight in Lemma 8 is also coupled with each other and is difficult to analyze. Then, due to the superposition of the previous two points, the relationship between the weights changing with time in the early stage of training (similar to Lemma 9) and the width and initialization of the neural network will be difficult to distinguish, which leads to the final result being complex, demanding and difficult to obtain a valid conclusion about width and initialization.

## G Societal impact

This is a theoretical work that explores the interplay of the width, the depth and the initialization of neural networks on their average robustness. Our goal is to obtain an in-depth understanding of the factors that affect the robustness. We do not focus on obtaining any state-of-the-art results in a

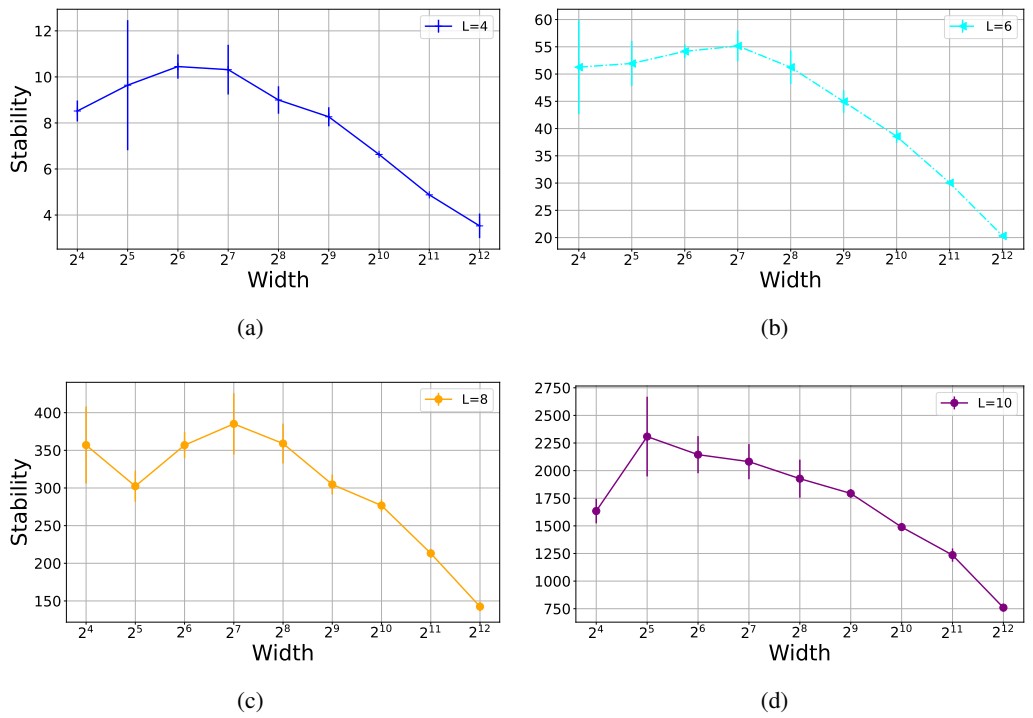

Figure 7: Relationship between the *perturbation stability* and width of CNN under He initialization for different depths of $L = 4, 6, 8$ and $10$. The stability values differ substantially across depths, which is more in line with the relationship between robustness and depth under He initialization in Theorem 1.

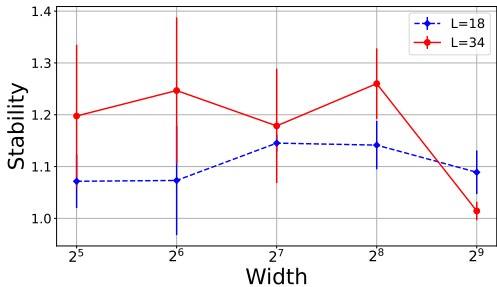

Figure 8: Relationship between the *perturbation stability* and width of ResNet-18 and ResNet-34.

particular task, which means there are other works that can be used for forming strong adversarial attacks and can be used with malicious intent.

Despite the theoretical nature of our work, we encourage researchers to further investigate the impact of robustness on the society. We expect robustness to have a key role into a world where neural networks are increasingly deployed into real-world applications.

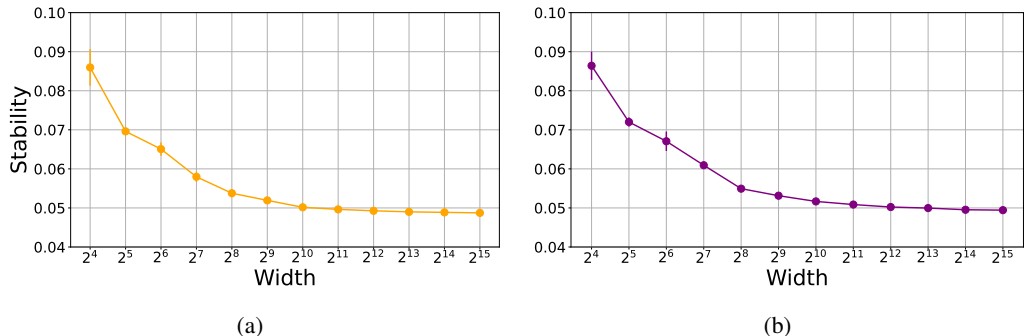

Figure 9: Influence of width of neural network on the perturbation stability under non-lazy training regime. (a) the variance of the initial weight is $\frac{1}{m^3}$. (b) the variance of the initial weight is $\frac{1}{m^4}$.

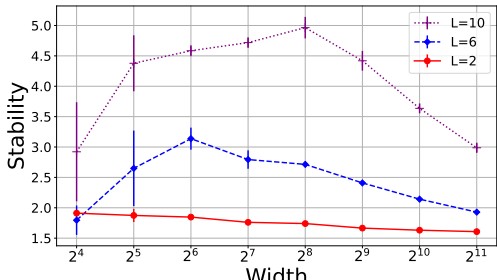

Figure 10: Relationship between the *perturbation stability* and depth of FCN under NTK initialization with different depths of $L = 2, 6$ and $10$.