# OpenReview forum: "Robustness in deep learning: The good (width), the bad (depth), and the ugly (initialization)"
_NeurIPS.cc/2022/Conference — NeurIPS 2022 Accept_

### Official Review · Reviewer_rKts · 2022-07-04

**Rating:** 7
**Confidence:** 4
**Soundness:** 3 good
**Presentation:** 4 excellent
**Contribution:** 4 excellent

**Summary:**

This work explores the (average) robustness of neural networks with respect to the width, the depth, and the initialization. After related work and problem settings, it derives theoretical bounds on average robustness (measured by perturbation stability, defined therein) for lazy (Theorem 1) and non-lazy (Theorem 3) training regimes. Interpreting these results novel conclusions are derived shedding light on contradictions in the previous literature outlined in Related Work. The experimental section is brief (FFN on MNIST dataset, accompanied by CNN experiments on CIFAR10 in Appendix) but well selected and clearly in support of theoretical results derived.

**Questions:**

Experiments use early stopping after 50 epochs. I understand that accuracy is not of concern given the scope of the work. Yet to confirm training has reached a somewhat reasonable level of loss, I suggest reporting on training/validation losses after early stopping is applied. It would further add to the clarity and relevance of the results.

**Limitations:**

Nothing to add to what is elaborated on in appendices F and G.

**Strengths And Weaknesses:**

To the best of the reviewer's knowledge, the work contains novel results and significantly extends the previous results on the robustness of neural networks. Experiments neatly presented in Figures, emphasize the main conclusions and suitably underpin the main narrative.

Significance is two-fold:
Firstly, it is due to improving previous perturbation stability bounds by Theorem 1 (and 3 for non-lazy training regime), neatly captured in Table 1. making the difference obvious.
The second novel result and perhaps evenly impactful is due to shedding light on previously contradicting results of the role of width related to robustness (discussed in the Related Works section as well). This is done by interpreting Theorem 1 (average perturbation stability upper bound for lazy training) and revealing a phase-transition-like phenomenon occurring in average stability w.r.t. width under a lazy training regime! This was only possible due to the less vacuous result of Theorem 1, compared to previous works.

Overall, the presented work is of high quality, clearly written, self-contained and wide in scope. Including a well-built argument and reasonably detailed proofs with preparatory lemmas (in appendix). The scope covers three different yet realistic initialization schemes (He, LeCun, and NTK) as well as lazy and non-lazy (limited to the two-layer net, but well-argued why in the appendix) training regimes.

Besides detailed proofs and additional experiments, the appendix includes a useful table of notation and analysis of the limitations of the work + comments on Societal Impact.

---

> ### Author Response · Authors · 2022-08-01
> **Response to reviewer rKts**
>
> We appreciate insightful feedback from the reviewer rKts. We address their main questions below:
>
> Q1: Reporting training/validation loses/accuracy after early stopping.
>
> A1: We have conducted our experiments on MLP with width = 32, depth = 4 and width = 64, depth = 8 for longer training (200 epochs). The experimental results (added in Appendix E.2 of the paper and displayed on the anonymous website https://imgur.com/a/faQwDsW) indicate that the loss and accuracy of the neural network almost do not change from the 50th epoch to the 200th epoch. Such results verify the rationality of our choice to run 50 epochs.
>
> Q2: Brief but well-selected numerical results.
>
> A2: According to your suggestions, we extend our results to some popular networks, e.g., ResNet. We have conducted our robustness experiments on ResNet-18 and ResNet-34 with different widths (from 32 to 512). The experimental results show a similar pattern to MLP, e.g., the existence of phase transition on the robustness. We listed the experimental results on stability as below:
>
> |Width|ResNet-18|ResNet-34|
> |---|---|---|
> |32|1.072 $\pm$ 0.052|1.198 $\pm$ 0.137|
> |64|1.073 $\pm$ 0.105|1.247 $\pm$ 0.141|
> |128|1.145 $\pm$ 0.016|1.179 $\pm$ 0.110|
> |256|1.141 $\pm$ 0.047|1.260 $\pm$ 0.068|
> |512|1.089 $\pm$ 0.042|1.014 $\pm$ 0.018|
>
> Thank you for the suggestion: The results coincide with our theoretical predictions and confirm our original validation, which strengthens our work. A detailed discussion on these results is shown in Appendix E.6 of the revised version of the paper.
>
> The updates are marked in red in the updated paper for convenience. If the reviewer has any remaining questions, we would be happy to elaborate further.

---

> > ### Author Response · Authors · 2022-08-08
> > **Any remaining questions from reviewer rKts?**
> >
> > Dear reviewer rKts,
> >
> > We are thankful for your constructive feedback. We have added experiments on early stopping and ResNet to the revised version paper.
> >
> > We would like to check whether there are any additional questions we could answer. We appreciate the time and feedback of the reviewer since it enables us to improve the quality of our work and the clarity of our contributions.

---

> > > ### Comment · Reviewer_rKts · 2022-08-10
> > > **Authors' response is appreciated**
> > >
> > > Thank you for adding experimental results (including training epochs extensions). Such doing alleviated my remaining minor concerns raised in the review. I'm happy with a paper in its current (revised) form as it stands. Best of luck!

---

### Official Review · Reviewer_xjE9 · 2022-07-05

**Rating:** 7
**Confidence:** 4
**Soundness:** 4 excellent
**Presentation:** 3 good
**Contribution:** 3 good

**Summary:**

The paper analyzes perturbation stability of fully connected feed-forward ReLU networks with respect to the networks width, depth, and initialization for the NTK regime, and with respect to width in the "non-lazy" regime. The results indicate that for overparameterized networks, stability increases with the width and that depth can increase or decrease robustness depending on the initialization. Moreover, the paper provides a criterion under which a network will train in the non-lazy regime (i.e., with large changes to the weights). The theoretical results are supported by an empirical evaluation.

The paper is well-written, the results are significant and interesting. The theoretical results are properly interpreted and empirically tested. This makes them quite insightful. This is a good paper. I vote for acceptance.

**Questions:**

- please define average robustness directly or at least be more clear how you use the terms (in the introduction, you use average robustness and $\epsilon$-stability interchangeably, in the remark after Def. 2 you clarify that $\epsilon$-stability is an approximation to average robustness).
- Def. 2 is a repitition from the intro and could be avoided
- the interpretation of the results could be written a bit clearer, in particular by using only $\epsilon$-stability or average robustness, so that it is more intuitive what it means if it goes up or down (also, exacerbate is a bit ambiguous here, since it can mean "make worse" or "enlarge", which was a bit confusing to me, in particular since I was not sure which of the two would be good).
- Out of curiosity: the results in [2] seem to indicate that flatness of the loss surface wrt. a single layer (independent of the rest of the network) implies average robustness. Can these results be related, e.g., through impacts of width and depth on flatness?

**Limitations:**

Please add a brief discussion on the difference of perturbation stability in expectation (as in Def. 2), and the also often used stability wrt. all data points (cf. [1]).

[1] Schmidt, Ludwig, et al. "Adversarially robust generalization requires more data." Advances in neural information processing systems 31 (2018).

[2] Petzka, Henning, et al. "Relative flatness and generalization." Advances in Neural Information Processing Systems 34 (2021): 18420-18432.

**Strengths And Weaknesses:**

Strengths:
- insightful theoretical results
- empirical validation of theory
- in-depth interpretation of results
- reproducible (code is provided)

Weaknesses:
- some minor improvements in clarity are necessary

---

> ### Author Response · Authors · 2022-08-01
> **Response to reviewer xjE9**
>
> We are thankful to the reviewer xjE9 for their feedback. We respond below to the remarks:
>
> Q1: Clarity: definition of average robustness and ϵ-stability. Def. 2 is a repetition of the introduction.
>
> A1: According to your suggestion, we have provided a formal definition of the perturbation stability in the introduction and removed this repetition in Sec. 3.2 in the revised version, see lines 28-33. Besides, discussion on various robustness metrics remains in Sec. 3.2.
>
> Since our definition of perturbation stability takes the expectation to clean and adversarial data points, it is natural to introduce the concept of average robustness. To avoid ambiguity in this work, we use perturbation stability to represent the quantity and adopt (average) robustness for ease of description. We have clarified this in our revised version, see lines 35-37.
>
>
>
> Q2: Discussion on the difference between perturbation stability and robust classification error of [1].
>
> A2: We are thankful to the reviewer for pointing out [1]. The definition of robustness in [1]
>
> $\beta = \mathbb{P}_{(x,y)~\sim\mathcal{P}}[\exists x' \in \mathcal{B}(x):f(x')\neq y]$
>
> relies on the misclassified error of an adversarial data point under the $\ell_{\infty}$-norm perturbation set. While our perturbation stability
>
> $\mathscr{P}(f, \epsilon) = \mathbb{E}_{x,\hat{x}} \left \| \nabla_x f(x)^{\top}(x-\hat{x}) \right \|_2 \,,\quad \forall x \sim \mathcal{D}_X,~~ \hat{x} \sim \text{Unif}(\mathbb{B} (\epsilon, x))$
>
> measures the function value changes at the clean data point via Taylor expansion under the $\ell_2$-norm perturbation set. Besides, our metric can be used for both classification and regression.
> We have included a short discussion in the revised version, see lines 155-158.
>
>
> Q3: Out of curiosity: the results in [2] seem to indicate that flatness of the loss surface wrt. a single layer (independent of the rest of the network) implies average robustness. Can these results be related, e.g., through impacts of width and depth on flatness?
>
> A3: Intuitively, flatness heavily depends on the structure of neural network architectures (e.g., width and depth) as well as the loss landscape, depending on where the training is also initialized.
>
>
> On the one hand, the proposed feature robustness in [2] shares a similar spirit to our average robustness. Both of them evaluate the change in output/loss of a neural network under the original data perturbation (in our setting) and feature perturbation (in [2]). Furthermore, the feature robustness is approximated by the relative flatness around a local minimum of ERM, which provides a possible way to connect flatness and robustness.
>
> On the other hand, [2] also links relative flatness to generalization, as well as previous work [3, 4] that connects robustness and generalization. So it would be possible to make a concise connection between relative flatness and our average robustness. In this case, flatness could also be characterized by the interplay of network width, depth, and initialization.
>
> We have included a short discussion in the revised version, see lines 327-329.
>
>
> The updates are marked in red in the updated paper for convenience. If the reviewer has any remaining questions or concerns, we would be happy to elaborate further.
>
>
> ### References
>
>
> [1] Schmidt, Ludwig, Santurkar Shibani, Tsipras Dimitris, Talwar Kunal, and Mądry Aleksander. "Adversarially robust generalization requires more data." NeurIPS (2018).
>
> [2] Petzka, Henning, Kamp Michael, Adilova Linara, Sminchisescu Cristian, and Boley Mario. "Relative flatness and generalization." NeurIPS (2021).
>
> [3] Yin, Dong, Ramchandran Kannan, and Bartlett Peter. "Rademacher Complexity for Adversarially Robust Generalization." International conference on machine learning, 2019.
>
> [4] Attias, Idan, Kontorovich Aryeh, and Mansour Yishay. "Improved Generalization Bounds for Robust Learning" Proceedings of the 30th International Conference on Algorithmic Learning Theory (2019).

---

> > ### Comment · Reviewer_xjE9 · 2022-08-05
> > **Response to the authors**
> >
> > Dear authors,
> >
> > **Q1:** Thank you for your response and the modifications of the manuscript. The definition of $\epsilon$-perturbation stability is indeed much clearer, as well as its relation to average robustness. A minor comment: "... it is natural to describe average robustness of neurar network." -> "... it is natural to describe *the* average robustness of *a* neural network.".
> >
> > **Q3:** This was really more out of curiosity, so feel free to leave the discussion out of the manuscript. I was just really interested, since it seems that your results allow to bound feature robustness, as well - without using flatness. So combining those two works could yield a generalization bound based on width, depth and initialization.
> >
> > Cheers

---

> > > ### Author Response · Authors · 2022-08-06
> > > **Thanks for your response**
> > >
> > > Dear reviewer xjE9,
> > >
> > > We are thankful to the reviewer xjE9 for the new feedback. According to your suggestion, we have fixed these two issues and we have uploaded a new version of the paper.
> > >
> > > Best,
> > >
> > > Authors

---

### Official Review · Reviewer_akix · 2022-07-10

**Rating:** 5
**Confidence:** 2
**Soundness:** 2 fair
**Presentation:** 2 fair
**Contribution:** 2 fair

**Summary:**

This paper discusses interplay of network width, depth and initialization on their average robustness with theoretical bounds. The authors show that width hurts robustness in the under-parameterized setting and improves robustness in the over-parameterized setting. Effect of depth depends on network intialization and training mode. In lazy training regime, depth helps robustness with LeCun initialization but hurts robustness with NTK and He-initialization. Under non-lazy training regime they demonstrate width helps robustness of a two-layer ReLU network.

**Questions:**

1. Some statements in the paper is confusing: definition 2 calls the mesaurement perturbation stability while inidicating a small measurement implies more robust function: so low stability means high robustness ? Same goes for the remak in 4.1
2. In numerical evidence section 5.4, validation for depth only showed 2 choices of depth which could take more choices to be more convincing. NTK is also missing for validation.

**Limitations:**

Theoretical assumptions are properly stated and explained.

**Strengths And Weaknesses:**

Strengths:
* Originality: The theoretical bounds derived by the authors studying network robustness wrt. depth, width and initialization are original.
* Quality: The theoretical derivation are well established with analysis and numerical evidence.

Weakness:
* Clarity: Some statements in the paper is confusing: definition 2 calls the mesaurement perturbation stability while inidicating a small measurement implies more robust function: so low stability means high robustness ? Same goes for the remak in 4.1
* Significance: The authors discussed the interplay of width, depth and initialization on network robustness and given the theoretical derivation found the effect on robustness has a change of phase as network goes from under-parameterized to over-parameterized which may settle some contradiction in previous literature, making a fair significance contribution.

---

> ### Author Response · Authors · 2022-08-01
> **Response to reviewer akix**
>
> We are thankful to the reviewer akix for their feedback. We respond below to the remarks:
>
> Q1:  Does low stability mean improved robustness?
>
> A1: Exactly. A lower stability value indicates improved robustness. In fact, the perturbation stability and the robustness of the neural network admit a negative correlation, which has been discussed/studied in sec. 3.2.
> We have clarified this in our revised version, see lines 35-37.
>
>
> Q2: May settle some contradiction in previous literature, making a fair significant contribution.
>
> A2: We are surprised about your significant remark, which reads positive, under the weakness. Other reviewers also recognize this significance as a positive towards our work.  Perhaps we misunderstood your remark but we will add the following:
>
> We provide a sharper analysis of the relationship between the robustness and overparameterization (involving width and depth) under different initializations and support the theory via empirical validations.
>
> Here we briefly discuss why our proof is able to obtain the refinements when compared to [1]. We take one example of how to bound the stability.
>
> The key technique in the proof of the main theorem in [1; Theorem 1] first estimates the Lipchitz constant via the product of the norm of the weight matrices in every layer $\prod_{j=1}^{n}  \lVert \theta_j  \rVert_2$, and then applies the classical result on maximum eigenvalue estimation of random Gaussian matrix [2] to bound the norm of weight matrices $  \lVert \theta_j  \rVert_2$ in each layer as follows:
>
> $L(f_\theta) \leq \prod_{j=1}^{n} \lVert \theta_j \rVert_2 \leq \prod_{j=1}^{n} \beta_j(\sqrt{m_j}+\sqrt{m_{j-1}}).$
>
> Nevertheless, this approach directly employs a uniform upper bound on the norm of weight matrices in all the layers, and thus ignores the relationship between the different layers of the neural network. In our proof, the coupling relationship between the different layers is preserved, and every weight is considered together,
>
> $E_{x,\hat{x},W} \lVert W_{L} D_{L-1} \cdots D_{1} W_{1}(x-\hat{x})\rVert_2^2 \leq m o\beta_1^2\beta_L^2\gamma^{2(L-2)}\epsilon^2/2.$
>
> Therefore, by using the special property of lazy-training
>
> $E_{x,\hat{x},W} \lVert  \nabla_{x} f(x)^{\top}(x-\hat{x}) - W_L D_{L-1}\cdots D_1 W_1(x-\hat{x})\rVert_2 \leq  \Theta\bigg(\epsilon\gamma^{L-2}\sqrt{\pi L^3m^2\beta_1^2\beta_L^2/8} \ e^{-m/L^3} \bigg),$
>
> a tighter upper bound can be obtained. (See details in Lemma 4 and Lemma 5 in Appendix B.1)
>
>
> Q3: Experimental validation for different depths.
>
> A3: We provided numerical results with depth $L \in {2, 4, 6, 8, 10}$ in Fig. 6 (Appendix E.2 of the supplementary). We have moved this result to the main paper by updating Fig. 5 according to the reviewer’s suggestions. The same figure can be found on https://imgur.com/a/AjKDk3k for convenience.
>
>
> Q4: Experimental validation for NTK initialization.
>
> A4: We are thankful to the reviewer for their suggestion. We originally did not include the NTK since practical neural networks are not frequently initialized with the NTK, however, we understand that it adds value. The experimental results (added in Appendix E.5 of the paper and displayed on https://imgur.com/a/7BIMMZQ) verify our theoretical results about the relationship of width and depth with robustness under the NTK initialization. That is, robustness first becomes worse in the under-parameterized setting then gets better, and finally tends to be a constant in highly over-parameterized regions and depth hurts robustness in a polynomial order.
>
>
> The updates are marked in red in the updated paper for convenience. We hope that our response has addressed the concerns of the reviewer on our paper; we would be happy to elaborate further if something remains unclear.
>
> ### References
>
> [1] Huang, Hanxun, Wang Yisen, Erfani Sarah, Gu Quanquan, Bailey James, and Ma Xingjun. "Exploring Architectural Ingredients of Adversarially Robust Deep Neural Networks." NeurIPS (2021).
>
> [2] Roman, Vershynin. "High-Dimensional Probability: An Introduction with Applications in Data Science." Cambridge University Press, 2018.

---

> > ### Author Response · Authors · 2022-08-08
> > **Any remaining questions from reviewer akix?**
> >
> > Dear reviewer akix,
> >
> > We welcome your feedback and thoughtful questions. We revised the paper to clarify the relationship between robustness and stability, discussed in detail how this paper settled some contradictions in previous literature and rearranged the experiment section by adding some experiments on NTK initialization.
> >
> > Since the discussion period is closing soon, we would like to check if the reviewer has been covered by our responses and the improvements in our manuscript.

---

### Meta-Review · Area_Chair_5zSA · 2022-08-26

**Recommendation:** Accept
**Confidence:** Certain

**Metareview:**

This paper studies an average notion of robustness and its relation to the depth, the width, and the initialization scheme. The reviewers found the theoretical results insightful and novel, the empirical results thorough, and the paper overall well-written and of high quality. Therefore, I recommend acceptance.

**Award:**

No

---

### Decision · Program_Chairs · 2022-09-14

Accept